# GsMTx-4 inhibits the exercise pressor reflex and the muscle mechanoreflex primarily through TRPC inhibition

Guillaume P. Ducrocq[1,2] , Laura Anselmi[1] , Kristen Brandt[1], Jianli Wang[3], Victor Ruiz Velasco[1,4] and Marc P. Kaufman[1]

[1] *Heart and Vascular Institute, Penn State College of Medicine, Hershey, PA, USA*
[2] *Mitochondrial, Oxidative Stress and Muscular Protection Laboratory (UR3072), Faculty of Medicine, University of Strasbourg, Strasbourg, France*
[3] *Department of Radiology, Penn State College of Medicine, Hershey, PA, USA*
[4] *Department of Anesthesiology and Perioperative Medicine, Penn State College of Medicine, Hershey, Pennsylvania, USA*

Handling Editors: Vaughan Macefield & Mathew Piasecki

The peer review history is available in the Supporting Information section of this article (https://doi.org/10.1113/JP289092#support-information-section).

**Abstract figure legend** In decerebrated rats, injection of the TRPC6 channels antagonist (SAR7334) in the arterial supply of the triceps surae muscles was followed by injection of the TRPC6 and Piezo 2 channels antagonist (GsMTx-4) to isolate the role of Piezo 2 in evoking the muscle mechanoreflex and exercise pressor reflex. The exercise pressor reflex was evoked by static contraction or the muscle mechanoreflex was evoked by calcaneal tendon stretch. Renal nerve sympathetic activity, muscle tension, blood pressure and blood flow were recorded. In Chinese hamster ovary cells expressing Piezo 2 channels, patch-clamp experiments revealed that SAR7334 did not inhibit the inward current evoked by mechanical stretch of the cell membrane by hypotonic solution.

Artificial intelligence-generated content (AIGC): No content has been generated by an artificial intelligence.

**Abstract**  Evidence suggests that Piezo 2 and TRPC6 channels play important roles in evoking the mechanical component of the exercise pressor reflex. However, the pharmacological tools used in previous studies, namely GsMTx-4 (Piezo 2) and SAR7334 (TRPC6), have potential overlapping effects. GsMTx-4, in particular, inhibits TRPC6 channels in addition to Piezo 2. Consequently, we determined *in vivo* the initial and combined effects of serial injection of GsMTx-4 and SAR7334 in evoking the mechanical component of the exercise pressor reflex in male and female decerebrated rats. In addition, in heterologous cells expressing Piezo 2 channels, we determined the effect of SAR7334 on the inward current evoked by membrane stretch. The exercise pressor reflex and the mechanoreflex were evoked by statically contracting and passively stretching the triceps surae muscles before and after injection of GsMTx-4 ($51 \pm 8\mu\text{M}$) followed by SAR7334 ($51 \pm 8\mu\text{M}$; $n = 8$–9) or after injection of SAR7334 ($51 \pm 8\mu\text{M}$) followed by GsMTx-4 ($51 \pm 8\mu\text{M}$; $n = 8$). *In vivo*, GsMTx-4 and SAR7334 inhibited the pressor and sympathetic nerve responses to passive stretch and static contraction when injected first. When GsMTx-4 was injected secondary to SAR7334, no reduction in blood pressure and sympathetic responses to static contraction or passive stretch was observed. *In vitro*, 1 μM of SAR7334 increased the inward current evoked by membrane stretch in heterologous cells expressing Piezo 2 ($n = 6$). In contrast, 4 μM increased the inward current in two cells, did not change the current in one cell and decreased the current in three cells. Our findings suggest that GsMTx-4 in our *in vivo* experiments antagonized TRPC as well as Piezo 2 channels.

(Received 15 April 2025; accepted after revision 31 July 2025; first published online 24 August 2025)

**Corresponding author** Guillaume P. Ducrocq: Heart and Vascular Institute, Penn State College of Medicine, Hershey, Pennsylvania, USA.    Email: gducrocq@unistra.fr

## Key points

- GsMTx-4 blocks both Piezo 2 channels and TRPC6 raising the possibility that its effects on the exercise pressor reflex and the mechanoreflex is mediated through TRPC6 inhibition.
- We determined the effects of GsMTx-4 alone, and after pre-treatment with a TRPC6 antagonist (SAR7334), to elucidate the potential overlapping effects of the drug on the exercise pressor reflex and the mechanoreflex.
- GsMTx-4 alone significantly inhibited the pressor and sympathetic responses to static contraction and passive stretch.
- However, when GsMTx-4 was injected after pre-treatment with a TRPC6 antagonist, the effects of GsMTx-4 were abolished.
- These results demonstrate that the effects of GsMTx-4 are mediated through TRPC6 channel inhibition and that the role played by Piezo 2 was previously overestimated.

## Introduction

The haemodynamic response to exercise is determined by coordinated neural mechanisms that aim to improve and redistribute blood flow towards the exercising muscles (Fisher et al., 2015). One of the major contributors to this response is the exercise pressor reflex which is evoked by the contraction-induced stimulation of thinly

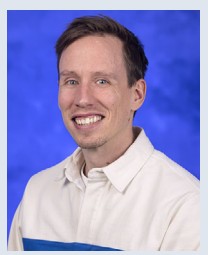

**Guillaume Ducrocq** is an Associate Professor at the Faculty of Sports Sciences, University of Strasbourg (France). His research focuses on the role played by group III and IV afferent fibres in regulating the cardiovascular and neuromuscular response to exercise in health and disease. His experiments are conducted in humans, animals and cells using an integrative approach.

myelinated (group III) and unmyelinated (group IV) afferent fibres (Coote & Perez-Gonzalez, 1970; Coote et al., 1971; McCloskey & Mitchell, 1972). The central endings of group III and IV afferents synapse onto neurons in lamina I, II and V of the dorsal horn of the spinal cord (Craig & Mense, 1983; Mense & Craig, 1988) and then project to the brainstem to evoke sympathetic activation (Victor et al., 1989) and vagal withdrawal (Al-Ani et al., 1997; McMahon & McWilliam, 1992). This in turn increases cardiac rate and contractility, constricts peripheral arteries and increases systemic blood pressure (Fisher et al., 2015).

Group III–IV afferent fibres are activated by mechanical distortion of their receptive fields (Kaufman et al., 1983; Mense & Stahnke, 1983; Paintal, 1960) and/or increased concentrations of muscle metabolites (Hoheisel et al., 2004; Kaufman & Rybicki, 1987; Mense & Meyer, 1985; Rotto & Kaufman, 1988). The molecular mechanisms that elicit the transduction of metabolic stimuli by group III–IV afferents have drawn considerable attention. For example, numerous studies showed that increased intramuscular concentrations of metabolic or inflammatory by-products are detected by a variety of receptors from the purinergic, endoperoxide and transient receptor potential (TRP) families (Butenas et al., 2024; Koba et al., 2011; Yamauchi et al., 2013). Less attention, however, has been devoted to determining the molecular mechanisms that are responsible for evoking the mechanical component of the exercise pressor reflex. As of today, three receptors have been identified as contributors; namely, TRP vanilloid 4 (Fukazawa et al., 2023), Piezo 2 (Copp et al., 2016a) and TRP canonical 6 (TRPC6 (Ducrocq et al., 2025)). Using a dose–response approach and two structurally different antagonists, we found that the latter contributed at least 50% of the pressor response evoked by the mechanoreflex in decerebrated rats (Ducrocq et al., 2025). Similarly, Copp et al. (2016a) found that blocking Piezo channels with GsMTx-4 (∼25 μM) also reduced the pressor and sympathetic nerve response to the mechanoreflex by 50%. This effect was mostly attributed to Piezo 2 rather than to Piezo 1 as the latter was minimally expressed in rats' dorsal root ganglion cells innervating the hindlimb. The effects of GsMTx-4 in blocking the mechanoreflex were later replicated in a rat model of Type I diabetes, establishing further the role played by Piezo 2 channels (Grotle et al., 2021).

Experiments on cell cultures found that GsMTx-4 not only blocks Piezo channels, but also blocks TRPC6 and TRPC1 channels (Alessandri-Haber et al., 2009; Spassova et al., 2006). Specifically, Spassova et al (2006) found that the inward current of HEK293 cells expressing TRPC6 channels in response to membrane stretch was almost fully abolished following exposure of these cells to 5 μM of GsMTx-4. These results raise the possibility that the inhibitory effects of GsMTx-4 on the exercise pressor reflex and the mechanoreflex previously documented were not entirely the result of Piezo 2 inhibition but also, at least partly, the result of TRPC6 inhibition. Alternatively, it is unknown whether TRPC6 antagonists, such as SAR7334, act on Piezo channels. Consequently, more data are needed to elucidate the specific contribution of Piezo 2 and TRPC6 channels in evoking the mechanical component of the exercise pressor reflex.

This need prompted us to determine: (1) *in vivo*, the effects of serial injections of GsMTx-4 and SAR7334 on the pressor and sympathetic responses to both static contraction and passive stretch; and (2) *in vitro*, the effects of SAR7334, a TRPC6 antagonist, on the function of Piezo 2 in cells expressing the Piezo 2 channel. We hypothesized that the administration of either SAR7334 or GsMTx-4 alone would attenuate these pressor and sympathetic responses. Additionally, we proposed that pre-treatment with SAR7334, which blocks TRPC6, would diminish the inhibitory effects of a subsequent GsMTx-4 injection. Conversely, we hypothesized that administering GsMTx-4 prior to SAR7334 would reduce the effectiveness of SAR7334's inhibitory actions.

## Material and methods

### Ethical approval

The Institutional Care and Use Committee of the Pennsylvania State University College of Medicine approved all of the procedures performed in the experiments to be described (IACUC: PRAM201647038). The authors understood and conformed to the ethical ARRIVE guidelines for animal use in research.

### Animal characteristics, wellness and sample size

Experiments were conducted at constant room air temperature (21°C) on male and female Sprague-Dawley rats (Charles River laboratory), weighing ∼250–450 g. Vaginal smears were collected from female rats and analysed with a commercially available kit to verify that the experiments were conducted during the diestrus phase of their oestrus cycle (i.e. the phase with the lowest concentration of female-specific sex hormones) (Koba et al., 2012). Rats were housed within the central animal facility of the Pennsylvania State University College of Medicine, with access to food and water *ad libitum*, and were exposed to a 50:50 light/dark cycle. All attempts were made to minimize animal discomfort and pain. An equal number of male and female was used in our experiments unless specified. The minimal sample size required for our experiments was estimated with G*Power (RRID:SCR_01 3726; 3.1.9.6, Dusseldorf, Germany) as 7, considering the effect size of SAR7334 previously reported

(Ducrocq et al., 2025; partial $\eta^2 = 0.526$ for contractions and partial $\eta^2 = 0.494$ for stretch experiments) an $\alpha$ level of 0.05 and statistical power $(1-\beta)$ of 0.95.

### Experimental protocols for the effects of SAR7334 on Piezo 2 channels *in vitro*

In this set of experiments, Chinese hamster ovary (CHO) cells were plated on 35 mm dishes at 80,000 cells/dish 18 hr prior to transfection with Piezo 2 and the enhanced green fluorescent protein (pEGFP) cDNA constructs. The cells were kept in a humidified incubator and grown on Dulbecco's modified Eagle's medium (DMEM) which was supplemented with 10% fetal bovine serum, 1% penicillin–streptomycin, 1% glutamine and 1% non-essential amino acids. The Piezo 2 cDNA construct was purchased from Origene (Catalogue No. MR226955). On the day of transfection, 12–20 µg of Piezo 2 cDNA was complexed with 2.5 µl of lipofectamine 2000 (Thermo-Fisher Inc., Waltham, MA) and the pEGFP construct (1 µg) was complexed with 2.5 µl of lipofectamine 2000 for 15 min. After the complexing period, both constructs were mixed and then added to the CHO cells which were plated with 1000 µl of DMEM (non-supplemented). The cells were transfected for a period of 5 h. Thereafter, they were rinsed twice with DMEM (supplemented) and then placed back in the incubator overnight.

### Electrophysiology and data analysis

Piezo 2 currents in GFP-expressing CHO cells were recorded at room temperature using the whole-cell variant of the patch-clamp technique with an Axopatch 200B amplifier (Molecular Devices, San Jose, CA) and an ITC-18 data acquisition interface (HEKA Instruments). The patch pipettes were pulled from borosilicate glass capillaries (No. 8250, King Precision Glass, Claremont, CA) on a P-97 Flaming-Brown micropipette puller (Sutter Instruments, Novato, CA), and fire polished on a microforge. Data acquisition was performed with the custom-designed software (F6) developed by Stephen R. Ikeda (National Institute on Alcohol Abuse and Alcoholism). The Piezo 2 currents were filtered at 1–2 kHz (-3 dB) and digitized at 2–5 kHz with a four-pole low-pass Bessel filter.

The external (normal) solution consisted of (in mM): 137 NaCl, 5.9 KCl, 1.8 CaCl$_2$, 1.2 MgCl$_2$, 14 glucose and 10 HEPES (pH 7.40 with NaOH), while the hypotonic solution consisted of: 90 NaCl, 5 KCl, 2.4 CaCl$_2$, 1.3 MgCl$_2$, 10 glucose and 10 HEPES (pH 7.40 with NaOH) as previously described (Zhao et al., 2022). The pipette solution consisted of: 80 mM *N*-methyl-D-glutamine, 20 mM tetraethylammonium hydroxide, 11 mM EGTA, 1 mM CaCl$_2$, 20 mM CsCl, 40 mM CsOH, 4 mM MgATP, 0.3 mM Na$_2$GTP and 14 mM TRIS Creatine PO$_4$ (pH 7.21). The Piezo 2 currents were acquired with a holding potential of −70 mV. The cells were exposed to the hypotonic solution for 5 s.

### Experimental protocols for reflex experiments

**General surgical procedure.** Each rat was anaesthetized initially by inhalation of 4% isoflurane in O$_2$. Once the corneal reflex was abolished and when pinching the hind paw did not produce a withdrawal reflex, the rat's trachea was cannulated, and its lungs were mechanically ventilated (model 683, Harvard Apparatus Inc., Holliston, MA). The concentration of isoflurane was reduced to 2% for the rest of the surgery. The left and right common carotid arteries and right jugular vein were cannulated (RPT040; Braintree Scientific, Inc., Braintree, MA) to record arterial blood pressure (P23XL; Gould-Statham Instruments Inc., Los Angeles, CA), draw arterial blood samples, and inject drugs into the systemic circulation, respectively. We cannulated the left superficial epigastric artery (SUBL-140, Braintree Scientific Inc., Braintree, MA, USA) which is a side branch of the femoral artery. A snare (2.0 silk suture) was placed around the femoral artery and vein ∼0.5–1 cm upstream from the superficial epigastric artery and vein. When tightened, the snare partially trapped the solution in the hindlimb circulation. The popliteal artery was dissected and exposed to place a flow probe and record popliteal blood flow during the manoeuvres (0.5PSB and TS420; Transonic Systems, Ithaca, NY). Vascular conductance was calculated by the following formula: conductance (ml min$^{-1}$ 100 mmHg$^{-1}$) = 100 × blood flow (ml min$^{-1}$)/mean arterial pressure (mmHg) (Limberg et al., 2020).

The head of the rat was secured in a Kopf stereotaxic unit. The hip and the left ankle were secured with metal clamps to prevent movement during the contraction or stretch procedures. The calcaneus bone was severed and the Achilles tendon was connected to a force transducer (FT10; Grass Instrument Co., Quincy, MA) and a rack and pinion device. The left tibial nerve was isolated and hooked with a bipolar stainless-steel electrode. Using a blunted spatula, we decerebrated the rat by sectioning the brain ∼1 mm rostral to the superior colliculus (Dobson & Harris, 2012; Smith et al., 2001). Isoflurane was then discontinued, and the lungs were ventilated with room air. Using a retroperitoneal approach, we opened the left side of the abdomen to expose the kidney and renal artery and vein. A branch of the renal nerve was dissected and hooked up to a bipolar platinum-iridium electrode (778000, A-M Systems) and connected to a high-impedance probe (HIP511; Grass Instrument Co., Quincy, MA) to record sympathetic renal nerve activity

(Victor et al., 1989). The electrode was secured on the nerve with silicone glue (Kwik-sil; WPI inc., Sarasota, FL, USA). Blood arterial $PO_2$ (100–150 mmHg), $PCO_2$ (35–40 mmHg) and $[HCO_3^-]$ (22–26 mmol $l^{-1}$) were kept within physiological range. Body temperature was maintained around 37°C using a heating lamp. At the end of the experiment, hexamethonium (0.5 ml; 20mg $ml^{-1}$, Sigma-Aldrich, St. Louis, MO) was injected intravenously to verify that the recorded nerve activity corresponded to post-ganglionic sympathetic activity and to quantify background noise (Smith et al., 2006). The decerebrated rats were then killed by intravenous injection of a super-saturated KCl solution.

**Contraction of the triceps surae muscles.** All contractions in the present experiments were conducted isometrically. Baseline tension of the triceps surae muscles was set at 60–100 g and motor threshold was determined. The stimulator output (S88, Grass Instrument Co., Quincy, MA) was then set at a current intensity that corresponded to $1.5\times$ motor threshold. To contract the triceps surae muscles, the tibial nerve was stimulated for 30 s at 40 Hz (0.01 ms pulse duration). This manoeuvre reflexively increased arterial blood pressure. A second contraction was evoked after 10 min of recovery to verify reproducibility.

After 10 min of recovery, the snare around the femoral artery and vein was tightened, and then SAR7334 (7 µg $kg^{-1}$; 100 µl) or GsMTx-4 (equimolar and volume) was injected into the superficial epigastric arterial catheter. The snare was released 3 min after completing the injection and contraction was repeated 10 min after the drug injection. This protocol was repeated using the same timing but with the other antagonist injected (i.e. SAR7334 if GsMTx-4 was injected first or GsMTx-4 if SAR7334 was injected first). Care was taken that peak and integrated tension produced by the contractions were matched before and after treatments.

**Stretch of the triceps surae muscles.** Using a similar protocol and timing as that used for the contraction experiments, we determined the effects of SAR7334 and GsMTx-4 on the pressor response to passive stretch, a manoeuvre which presumably mimics the mechanical component of the exercise pressor reflex (Daniels et al., 2000; Stebbins et al., 1988). To avoid potential reflex muscle contractions evoked by stretch, the rats were paralysed by intravenous injection of pancuronium bromide (1 mg/ml, 200 µl) (Daniels et al., 2000). The triceps surae muscles were stretched for 30 s by turning a rack and pinion attached to the calcaneal tendon until a tension of ∼700 g was reached.

**Control for electrical activation of group III and IV axons.** To show that tibial nerve stimulation did not electrically activate the axons of the group III and IV afferent fibres during the contraction experiments, we paralysed the rat with pancuronium bromide (1 mg/ml, 200 µl; iv). The tibial nerve was then stimulated for 30 s at 40 Hz with the highest current used to evoke contraction (Daniels et al., 2000). If an increase in blood pressure was observed, the data were excluded from the dataset. No data were excluded based on this criterion.

**Control that the drug circulated to the triceps surae muscles.** To determine that injections into the hindlimb arterial circulation accessed the triceps surae muscles, we injected 0.2 ml of Evans Blue dye into the superficial epigastric artery catheter of each rat tested. We considered that the infusion circulated to the triceps surae muscles if they turned blue. If the colour of the muscles did not change, we excluded the data from the study. Two data points were excluded based on this criterion.

### Drug preparation

GsMTx-4 (1 mg; MedChemExpress) and SAR7334 hydrochloride (5 mg; MedChemExpress) were dissolved in 0.9% saline. The solutions were sonicated and vortexed. Care was taken that no flakes were visible. The stock solution for SAR7334 and GsMTx-4 corresponded to 160 µM and 80 µM in 100 µl, respectively. The concentration of SAR7334 that was injected was based on previous findings showing that 7 µg/kg of SAR7334 effectively inhibited the pressor response to static contraction and passive stretch in decerebrated rats (Ducrocq et al., 2025). An equimolar and volume of GsMTx-4 solution was prepared and injected to ensure that the same number of active molecules was present. The concentration of SAR7334 and GsMTx-4 that was used throughout the experiments was $51 \pm 8$ µM. To reach the different concentrations used during the experiments, the stock solutions were further diluted by adding the appropriate volume of saline.

### Data analysis

Renal sympathetic nerve activity was amplified (gain: $10 \times 1000$) and filtered (bandpass 30 Hz – 1 kHz) with a Grass P511 pre-amplifier (Grass Instrument Co., Quincy, MA). Renal sympathetic nerve activity, tension, developed by the contracting triceps surae muscles, and arterial blood pressure signals were amplified using Gould Universal amplifiers (Gould-Statham Instruments, Inc., CA). Except for renal sympathetic nerve activity, which was recorded at 10 kHz, all signals were recorded at 1 kHz using an A/D converter (Micro1401 mkII; Cambridge Electronic Design Limited, Cambridge, UK) and its

**Table 1. Baseline values for blood pressure, tension and blood flow during the contraction and stretch experiments**

| | Index | Units | Contraction | | | | Stretch | | | |
|---|---|---|---|---|---|---|---|---|---|---|
| | | | Pre | Post 1st Injection | Post 2nd Injection | *P*-value | Pre | Post 1st Injection | Post 2nd Injection | *P*-value |
| SAR7334 injected 1st | MAP | mmHg | 86 ± 21 | 84 ± 21 | 91 ± 15 | > 0.381 | 101 ± 32 | 111 ± 25 | 107 ± 28 | > 0.354 |
| | Tension | g | 85 ± 9 | 86 ± 8 | 88 ± 7 | > 0.846 | 68 ± 9 | 73 ± 7 | 74 ± 9 | > 0.228 |
| | Blood flow | ml min$^{-1}$ | 0.41 ± 0.13 | 0.37 ± 0.11 | 0.39 ± 0.11 | > 0.356 | | Not measured | | |
| GsMTx-4 injected 1st | MAP | mmHg | 81 ± 20 | 85 ± 21 | 82 ± 18 | > 0.337 | 105 ± 31 | 108 ± 33 | 109 ± 32 | > 0.847 |
| | Tension | g | 68 ± 20 | 73 ± 10 | 70 ± 15 | > 0.850 | 65 ± 11 | 76 ± 25 | 70 ± 15 | > 0.514 |
| | Blood flow | ml min$^{-1}$ | 0.40 ± 0.20 | 0.46 ± 0.22 | 0.44 ± 0.17 | > 0.219 | | Not measured | | |

*Note*: Data are presented as the means ± SD. *P*-values present the lowest *P*-value that was calculated between measurements. No difference was found between baseline measurements.
Abbreviation: MAP, mean arterial pressure.

associated commercially available software (Spike2, 7.20, RRID: SCR_000903; Cambridge Electronic Design Limited). Heart rate was calculated beat by beat from the arterial pressure pulse signal and expressed as beats per minute. Renal sympathetic nerve activity was rectified.

To determine the effects of static contraction or passive stretch on cardiovascular and sympathetic nerve function, we calculated the peak increase in blood pressure and the blood pressure index. The latter was calculated by integrating the area under the curve during the 30 s contraction or stretch period, and then subtracting from this value the area under the curve measured during the immediately preceding 30 s baseline period. The blood pressure index provides a measure of the entire pressor response, unlike the peak pressor response, which represents the instantaneous maximal value. Using a similar method, we calculated the change in peak tension, peak renal sympathetic nerve activity, peak blood flow and peak heart rate produced by the contraction or stretch as well as their integrated responses (i.e. the equivalent of the blood pressure index). The time course of blood pressure, tension, renal sympathetic nerve activity and blood flow to static contraction or passive stretch was plotted by averaging the mean signal every 2 s.

### Statistical analysis

Data are presented as the means ± SD. Because no effect of sex was found, male and female were pooled in one group. Using a Shapiro–Wilk's test, we determined whether our samples followed a normal distribution. The pre- to post-stimulus (i.e. static contraction, passive stretch or chemical injection) change in peak or integrated responses

as well as the changes in current in whole-cell patch-clamp experiments were evaluated using a two-tailed Student's paired *t* test. Differences between male and female rats were evaluated using a two-tailed Student's unpaired *t* test or its non-parametric equivalent Mann–Whitney's test. The difference between the pre- and post-blockade was evaluated with a one-way ANOVA with repeated measures or its non-parametric equivalent, the Friedman test. Differences in the time course of the different indices were evaluated using two-way ANOVA test with repeated measures (*measures × time*). The level of significance was set at $P < 0.05$. When a statistical difference was found with the ANOVA or Friedman tests, *post hoc* multiple-comparison analysis was performed using Tukey's honestly significant difference test or Dunn's test, respectively. Effect size was calculated using Cohen's *d* for Student's *t* test, partial $\eta^2$ ($_p\eta^2$) for ANOVAs (Cohen, 1977) or Kendall's W for the Friedman test (Tomczak & Tomczak, 2014). A Cohen's *d* or Kendall's W index for effect size was considered as small, medium or large when *d* was close to 0.2, 0.5 or 0.8, respectively (Cohen, 1977; Tomczak & Tomczak, 2014). A $_p\eta^2$ for effect size was considered as small, medium or large when $_p\eta^2$ was close to 0.02, 0.13 or 0.26, respectively (Cohen, 1977). Statistical analyses were conducted using Prism (RRID: SCR_0 02798; 10.3.0; GraphPad software; Boston, MA, USA).

### Results

#### Effects of SAR7334 and GsMTx-4 on baseline values

No effect of SAR7334 or GsMTx-4 was found on baseline values (Table 1).

### Effects of SAR7334 alone on the exercise pressor reflex and the mechanoreflex

Injecting SAR7334 into the circulation of the triceps surae muscles significantly inhibited the pressor, cardio-accelerator and sympathetic responses to both static contraction ($n = 8$; 4 females) and passive stretch (Figs 1–4; $n = 8$; 5 females). SAR7334 also decreased popliteal blood flow during static contraction without changing vascular conductance. SAR7334 reduced the integrated pressor responses to static contraction and to passive stretch by -60 ± 26% and -43 ± 25%, respectively (Figs 2, 4, 5). Likewise, the integrated renal sympathetic nerve activity responses to static contraction and to passive stretch were reduced by -61 ± 23% and -94 ± 65%, respectively. Analysis of the time courses revealed that the inhibition of blood pressure and renal sympathetic nerve activity had a latency of 6 s and 2 s, respectively.

### Effects of GsMTx-4 alone on the exercise pressor reflex and the mechanoreflex

Injecting GsMTx-4 into the circulation of the triceps surae muscles significantly inhibited the pressor, cardioaccelerator and sympathetic responses to static contraction ($n = 9$; five females) and to passive stretch (Figs 1–4; $n = 8$; four females). GsMTx-4 also decreased popliteal blood flow during static contraction without changing vascular conductance. GsMTx-4 reduced the integrated pressor responses to both static contraction and passive stretch by -38 ± 18% and -27 ± 23%, respectively (Figs 2; 4–5). In addition, the integrated renal sympathetic nerve activity responses to static contraction and passive stretch were reduced by -53 ± 24% and -72 ± 24%, respectively. Analysis of the time courses revealed that the inhibition of blood pressure and renal sympathetic nerve activity had latencies of 6 s and 2 s, respectively.

### Effects of GsMTx-4 when injected after SAR7334 on the exercise pressor reflex and on the mechanoreflex

When injected after SAR7334, GsMTx-4 had almost no effect on the pressor, cardioaccelerator and renal sympathetic nerve responses to either static contraction or passive stretch (Figs 1–4). Specifically, GsMTx-4 further decreased the integrated blood pressure response to both static contraction and passive stretch by only -3 ± 33% and -4 ± 11%, respectively (Fig. 5). This corresponds to a 92% and 86% reduction of its effects compared with when GxMTx-4 was injected alone. Similarly, the effects of GsMTx-4 subsequent to SAR7334 on the integrated renal sympathetic nerve responses to static contraction and to passive stretch were -4 ± 29% and 1 ± 58%, respectively. This corresponds to a 93% and 100% reduction of its

effects compared with when it was when injected alone. No difference in peak or integrated tension was found between treatments.

### Effects of SAR7334 when injected after GsMTx-4 on the exercise pressor reflex and on the mechanoreflex

When injected after GsMTx-4, SAR7334 further decreased the integrated blood pressure responses to static contraction and to passive stretch by -24 ± 12% and -20 ± 15% (Figs 2, 4), respectively. These effects were -60% and -53% lower than its effects compared with when SAR7334 was injected alone. Renal sympathetic nerve activity was further decreased by the injection of SAR7334 subsequent to GsMTx-4 by -18 ± 22% during static contraction but by just -1 ± 51% during passive stretch. These effects were -64% and -99% lower than its effects compared with when injected alone. No difference in peak or integrated tension was found between treatments.

### Effects of sex on the pressor responses to stretch and contraction and on the inhibitory effects of SAR7334 and GsMTx-4

When all pre-injection responses to static contraction or passive stretch were pooled together to increase sample size and statistical power ($n$ ranged from 7 to 9 for each sex), a greater peak pressor response was found in male than in female rats (Table 2). In addition, the integrated pressor response to static contraction was greater in male than female rats, whereas it was not greater during passive stretch. In contrast, the integrated renal sympathetic nerve activity in response to passive stretch was significantly greater in female than male rats, whereas it was not greater during static contraction. No difference in the integrated tension response to contraction or stretch was found between sexes.

The effects of SAR7334 or GsMTx-4 on the integrated pressor and renal sympathetic nerve activity responses to static contraction and passive stretch were not different between male and female rats (Figs 2, 4, 5).

**Effects of the SAR7334 on stretch-activated Piezo 2 currents.** Figure 6 shows current traces obtained in a Piezo 2-expressing CHO cell in the presence of a hypo-osmotic solution (left). After a recovery period, the cell was exposed to a hypoosmotic solution of 1 μM of SAR7334 (right). It can be observed that the Piezo 2 current increased in the presence of SAR7334. In all cells tested, the peak current increased when exposed to a hypoosmotic solution in the presence of SAR7334 (1 μM; $n = 6$, Fig. 6). In contrast, exposure of Piezo 2-expressing CHO cells to 4 μM of SAR7334 produced an increase of the

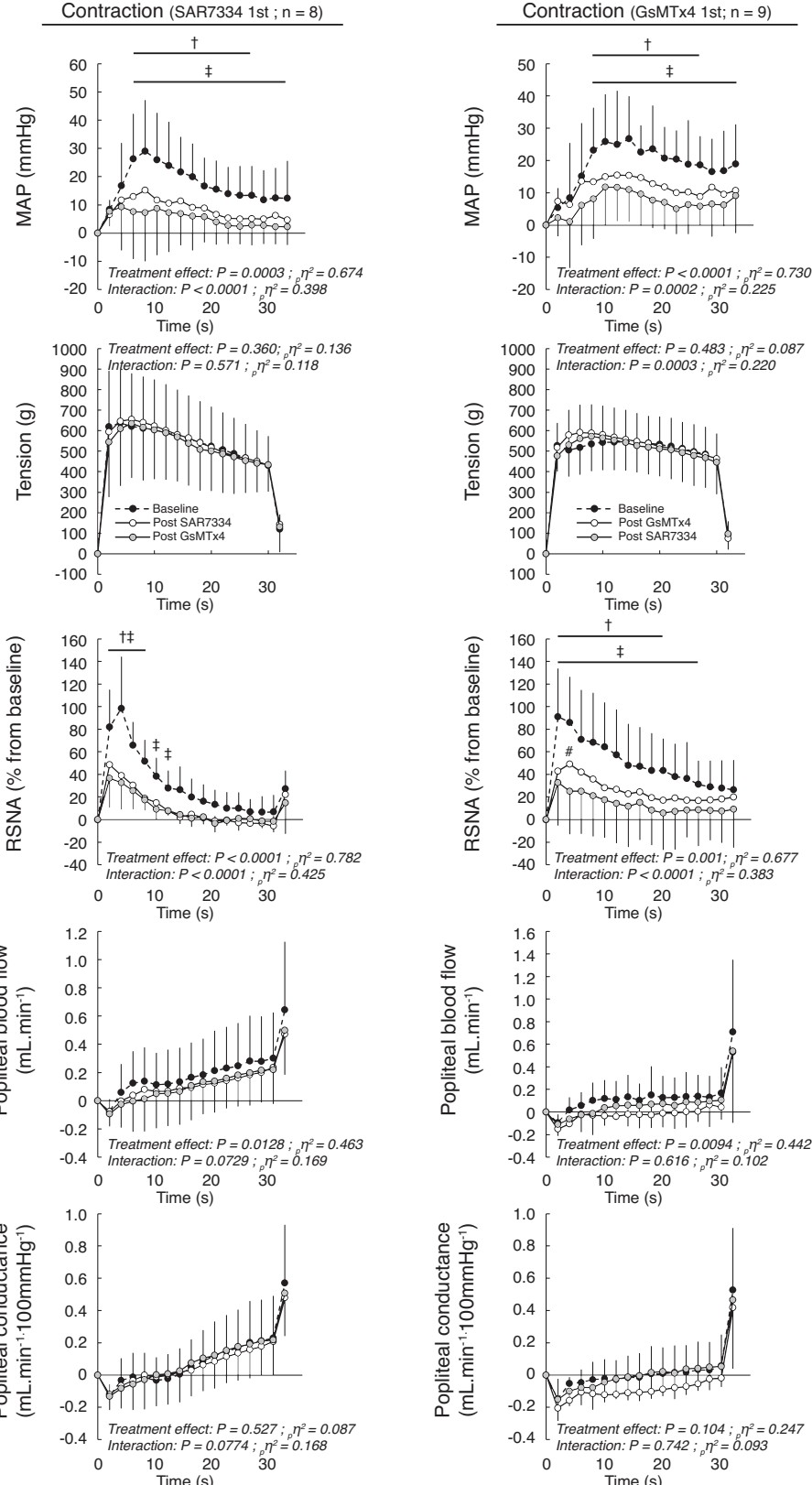

**Figure 1. Effects of serial injections of SAR7334 and GsMTx-4 on the pressor, tension, renal sympathetic, blood flow and vascular conductance responses to static contraction of the triceps surae muscles**
Data are presented as the mean ± SD changes over time in blood pressure, tension, RSNA, blood flow and vascular conductance induced by static contraction. The averaged time course includes 2 s of baseline, 30 s of contraction

and 2 s after the end of contraction. Contractions were evoked before, 10 min after the first injection, and 10 min after the second injection. Figures in the left panels represent the experiments during which SAR7334 (51 ± 8 μM in 100 μl) was injected first and GsMTx-4 (51 ± 8 μM in 100 μl) was injected second (*n* = 8). Figures in the right panels represent the experiments during which GsMTx-4 (51 ± 8 μM in 100 μl) was injected first and SAR7334 (51 ± 8 μM in 100 μl) was injected second (*n* = 9). Abbreviations: MAP, mean arterial pressure; RSNA, renal sympathetic nerve activity. †*P* < 0.05 between pre-blockade and first blockade. ‡*P* < 0.05 between pre-blockade and second blockade.

peak current evoked by membrane stretch in two cells, no change in one cell and a decrease (Fig. 6) in three cells. The difference in the current evoked by membrane stretch with SAR7334 compared with without SAR7334 did not reach statistical significance (*P* > 0.310; Fig. 6).

## Discussion

We investigated the initial and combined effects of GsMTx-4 and SAR7334 on the exercise pressor reflex and the mechanoreflex, as well as the effects of SAR7334 on the response to membrane stretch of Piezo 2 channels heterologously expressed in CHO cells. In our *in vivo* experiments, we found that either GsMTx-4 or SAR7334, injected alone, inhibited the pressor and sympathetic responses to both static contraction and to passive stretch. However, when GsMTx-4 was injected subsequent to SAR7334, the inhibitory effect of the former antagonist was greatly reduced. In contrast, when SAR7334 was injected subsequent to GsMTx-4, the inhibitory effect of the former antagonist was reduced, but only to a minor extent. Our findings suggest that GsMTx-4 in our *in vivo* experiments primarily antagonized TRPC6 channels rather than Piezo 2 channels. Contrary to the effect of GsMTx-4, which blocks several families of stretch-activated channels, including Piezo, TRPC6 and TRPC1 (Spassova et al., 2006), we found that 1 μM of SAR7334 increased Piezo 2 channel peak current in CHO cells, whereas 4 μM had a dual effect. That is, Piezo 2 currents were either blocked or enhanced.

### Inhibition of the exercise pressor reflex and the mechanoreflex by GsMTx-4 or SAR7334

Our findings showing that SAR7334 when injected alone drastically inhibited the pressor and sympathetic responses to both static contraction and passive stretch are consistent with previous work from our laboratory (Ducrocq et al., 2025). The approximate 60% and 40% respective reduction in the integrated blood pressure response to static contraction and to passive stretch by SAR7334 is comparable to the ~45% and 35% reduction previously reported (Ducrocq et al., 2025). The effects of SAR7334 on renal sympathetic nerve activity were not evaluated in our previous study (Ducrocq et al., 2025). Consequently, the findings of the present experiments extend our understanding of the effects of the drug on the

responses to contraction and to passive stretch showing a magnitude and latency that were similar to those induced by BI-749327, another TRPC6 antagonist (Ducrocq et al., 2025).

Consistent with the work of Copp et al. (2016a), we found that GsMTx-4 inhibited the pressor and sympathetic responses to passive stretch (Figs 3, 4). Analysis of the time course of the effects of GsMTx-4 revealed that the drug inhibited the pressor response from 6 s through to the end of the manoeuvre. In contrast, Copp et al. (2016a) found that the pressor response to passive stretch was inhibited during the first 5 s but not during the second half of the mechanoreflex. This discrepancy might be the result of the greater concentration of GsMTx-4 that we used compared with that used by Copp et al. (2016a) (~25 μM); our concentration might have blocked a greater number of receptors and/or had a longer effect. The greater concentration used in the present experiment was the consequence of our experimental design, which mandated that the concentration of GsMTx-4 matched the concentration of SAR7334 to ensure that the same number of molecules were injected.

In the present experiments, we found that GsMTx-4 inhibited the pressor and sympathetic responses to static contraction with a comparable magnitude and latency as those obtained from our passive stretch experiments. The fact that Copp et al. (2016a) found that the pressor response to passive stretch was inhibited only during the first 5 s of the manoeuvre led these authors to speculate that the effects of GsMTx-4 on mechanotransduction were predominantly involved when muscle tension changed dynamically, rather than when muscle tension was sustained. Evidence supporting that hypothesis was provided by the findings that the pressor responses to both intermittent contraction and stretch were inhibited throughout both manoeuvres, during which tension varied continuously (Copp et al., 2016a; Grotle et al., 2021; Sanderson et al., 2019). As a consequence, no data were previously available on the effects of GsMTx-4 during static contractions. In this context, direct comparison of our results obtained with static manoeuvres cannot be made with those obtained with intermittent manoeuvres (Copp et al., 2016a; Grotle et al., 2021; Sanderson et al., 2019). Further experiments are consequently needed to elucidate the effects of SAR7334 and GsMTx-4 in evoking the exercise pressor reflex and the mechanoreflex in decerebrate rats.

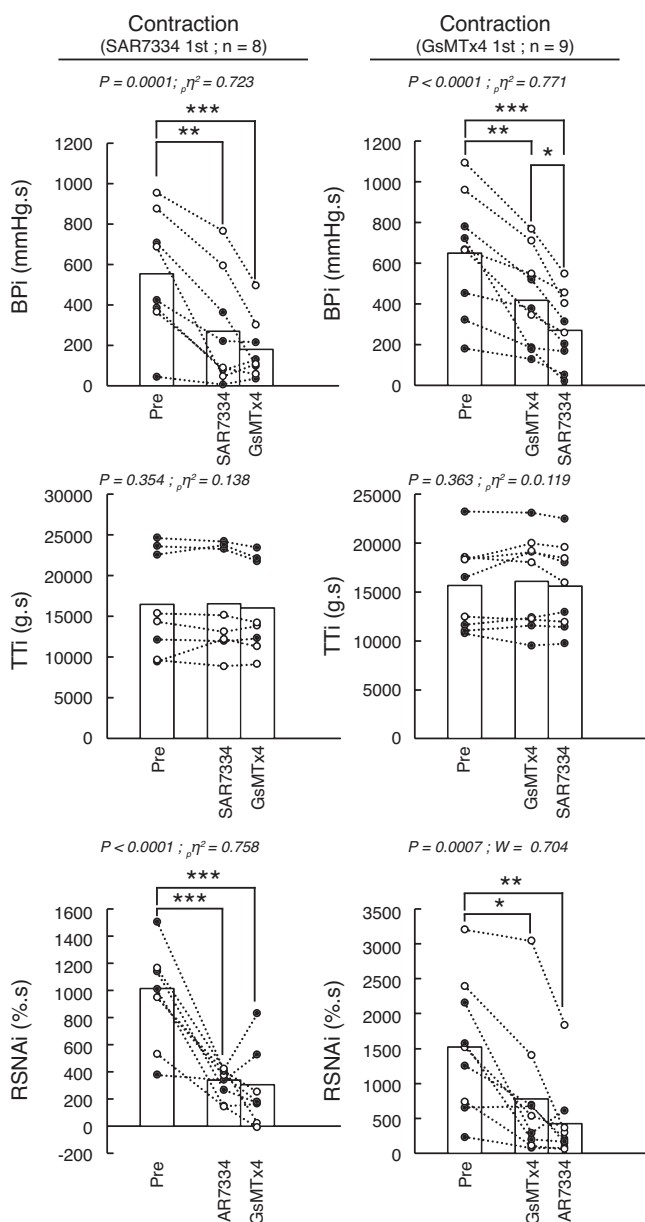

**Figure 2. Effects of serial injections of SAR7334 and GsMTx-4 on the integrated pressor, tension and renal sympathetic nerve activity responses to static contraction of the triceps surae muscles**

Data are presented as individual (males as open circles and females as closed circles) and group means (open bars) for the integrated changes evoked by static contraction. Contractions were evoked before, 10 min after the first injection, and 10 min after the second injection. Figures in the left panel represent the experiments during which SAR7334 (51 ± 8 μM in 100 μl) was injected first and GsMTx-4 (51 ± 8 μM in 100 μl) was injected second (*n* = 8). Figures in the right panel represent the experiments during which GsMTx-4 (51 ± 8 μM in 100 μl) was injected first and SAR7334 (51 ± 8 μM in 100 μl) was injected second (*n* = 9). Abbreviations: BPi, blood pressure index calculated as the integrated blood pressure response to contraction; TTi, tension time index calculated as the integrated tension response to contraction; RSNAi, integrated renal sympathetic nerve activity. ∗P < 0.05, ∗∗ P < 0.01, and ∗∗∗P < 0.001 between the corresponding data points.

**Table 2. Sex differences in the pressor response to static contraction and passive stretch before injecting SAR7334 or GsMTx-4**

| | | Contraction | | | Stretch | | |
|---|---|---|---|---|---|---|---|
| Index | Units | Male | Female | P-value; Cohen's d | Male | Female | P-value; Cohen's d |
| *n* | none | 8 | 9 | | 7 | 9 | |
| Peak MAP | mmHg | 54 ± 18 | 39 ± 10 | **P = 0.0152; d = 1.03** | 73 ± 15 | 50 ± 15 | **P = 0.0111; d = 1.48** |
| BPi | mmHg s | 782 ± 232 | 446 ± 251 | **P = 0.0120; d = 1.39** | 804 ± 354 | 978 ± 554 | P = 0.837; d = 0.372 |
| TTi | g s | 14,544 ± 3771 | 17,324 ± 6089 | P = 0.409; d = 0.549 | 15,255 ± 600 | 14,616 ± 1406 | P = 0.282; d = 0.591 |
| RSNAi | % s | 1428 ± 920 | 1100 ± 614 | P = 0.396; d = 0.420 | 324 ± 210 | 846 ± 375 | **P = 0.0054; d = 1.72** |

*Note*: Data are presented as the means ± SD. In bold font, *P* < 0.05.
Abbreviations: BPi, blood pressure index; MAP, mean arterial pressure; *n*, number of animals per analysis; RSNAi, integrated renal sympathetic nerve activity response; TTi, tension time index.

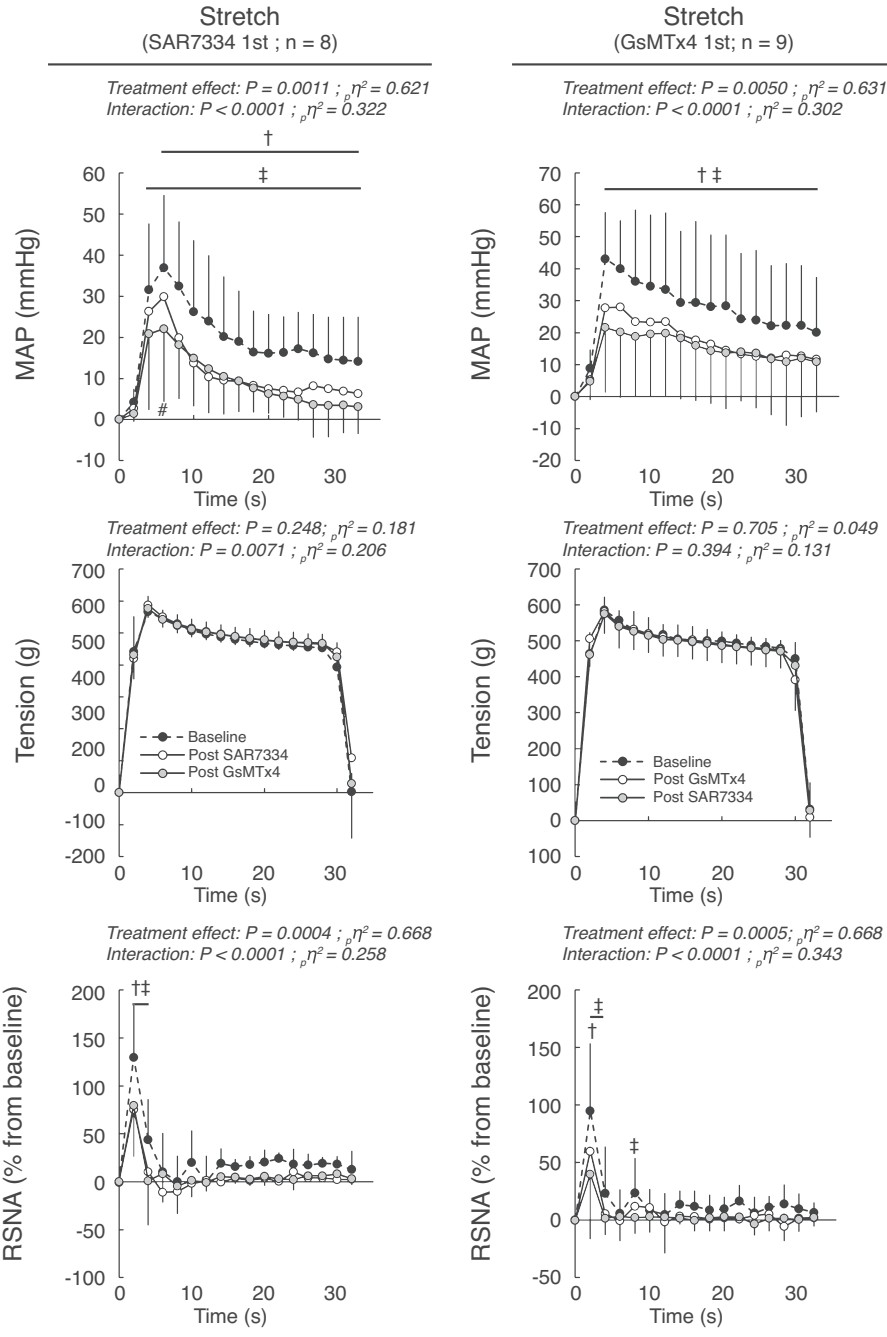

**Figure 3. Effects of serial injections of SAR7334 and GsMTx-4 on the pressor, tension, renal sympathetic, blood flow and vascular conductance responses to passive stretch of the triceps surae muscles**
Data are presented as the mean ± SD changes over time in blood pressure, tension and RSNA induced by passive stretch. The averaged time course includes 2 s of baseline, 30 s of contraction and 2 s after the end of stretch. Stretches were evoked before, 10 min after the first injection, and 10 min after the second injection. Figures in the left panels represent the experiments during which SAR7334 (51 ± 8 μM in 100 μl) was injected first and GsMTx-4 (51 ± 8 μM in 100 μl) was injected second (*n* = 8). Figures in the right panels represent the experiments during which GsMTx-4 (51 ± 8 μM in 100 μl) was injected first and SAR7334 (51 ± 8 μM in 100 μl) was injected second (*n* = 9). Abbreviations: MAP, mean arterial pressure; RSNA, renal sympathetic nerve activity. †*P* < 0.05 between pre-blockade and first blockade. ‡*P* < 0.05 between pre-blockade and second blockade. # *P* < 0.05 between the first and second blockade.

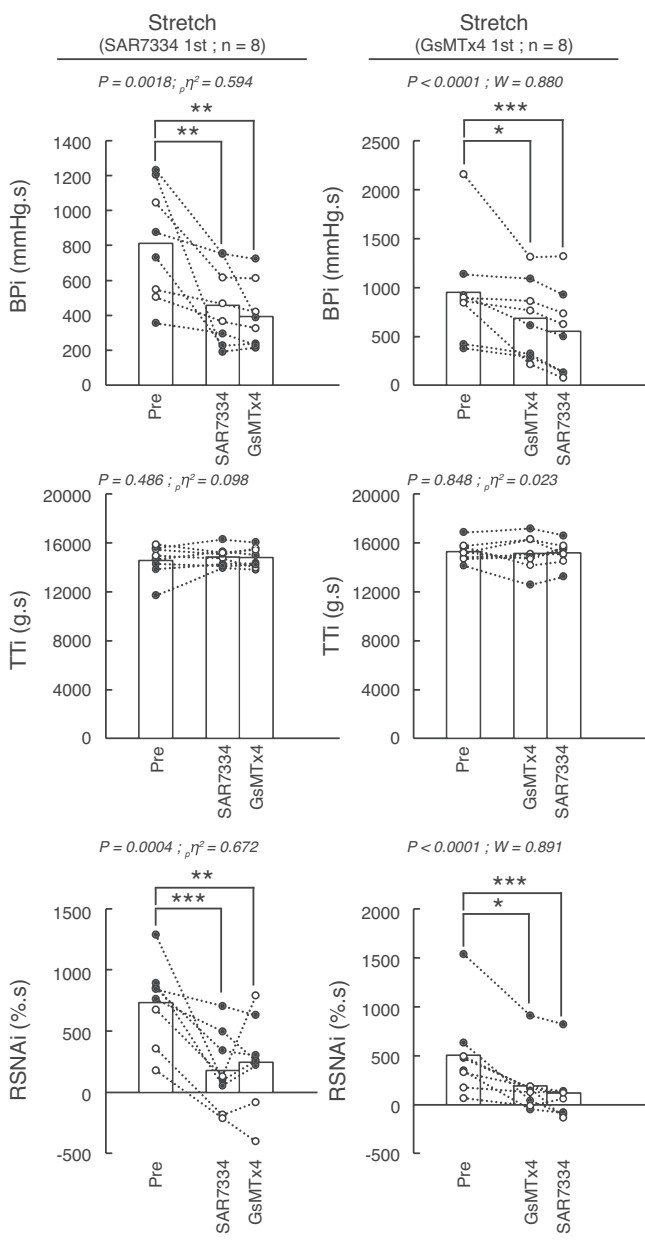

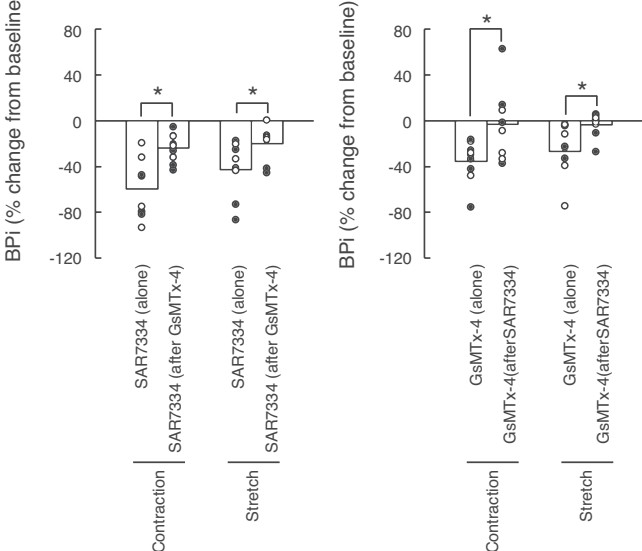

## Pre-inhibition of TRPC6 with SAR7334 nearly abolished the effects of GsMTx-4 during static contraction and passive stretch

To determine the overlapping effects of SAR7334 and GsMTx-4, we injected GsMTx-4 secondary to SAR7334 and evoked static contraction or passive stretch. No further reduction of the blood pressure and sympathetic responses to static contraction and passive stretch was found when GsMTx-4 was injected after SAR7334. These findings raise the possibility that the effects of GsMTx-4 previously reported during static stretch were, at least in part, the results of TRPC6 inhibition rather than Piezo 2 inhibition (Copp et al., 2016a, 2016b). Our results extend our understanding of the mechanism of mechano-transduction in health, but also potentially in various disease models. For example, the effects of GsMTx-4 have been investigated in rat models of peripheral artery disease and Type I diabetes(Copp et al., 2016b; Grotle et al., 2019). In particular, Copp et al. (2016b) found that GsMTx-4 had a greater effect on the blood pressure response to static

**Figure 4. Effects of serial injections of SAR7334 and GsMTx-4 on the integrated pressor, tension and renal sympathetic nerve activity responses to passive stretch of the triceps surae muscles**
Data are presented as individual (males as open circles and females as closed circles) and group means (open bars) for the integrated changes evoked by passive stretch. Stretches were evoked before, 10 min after the first injection, and 10 min after the second injection. Figures in the left panel represent the experiments during which SAR7334 (51 ± 8 μM in 100 μl) was injected first and GsMTx-4 (51 ± 8 μM in 100 μl) was injected second (*n* = 8). Figures in the right panel represent the experiments during which GsMTx-4 (51 ± 8 μM in 100 μl) was injected first and SAR7334 (51 ± 8 μM in 100 μl) was injected second (*n* = 9). Abbreviations: BPi, blood pressure index calculated as the integrated blood pressure response to contraction; TTi, tension time index calculated as the integrated tension response to stretch; RSNAi, integrated renal sympathetic nerve activity. ∗*P* < 0.05, ∗∗ *P* < 0.01, and ∗∗∗*P* < 0.001 between the corresponding data points.

**Figure 5. Percentage change from baseline of the integrated blood pressure response to static contraction and passive stretch following injection of SAR7334 and/or GsMTx-4**
Data are presented as individual (males as open circles and females as closed circles) and group means (open bars) for the percentage changes from baseline in the integrated blood pressure evoked by static contraction and passive stretch. Stretches and contractions were evoked before, 10 min after the first injection, and 10 min after the second injection. Figure in the left panel represents the effects of SAR7334 (51 ± 8 μM in 100 μl) alone and the effects of SAR7334 when injected after GsMTx-4 (51 ± 8 μM in 100 μl). Figure in the right panel represents the effects of GsMTx-4 (51 ± 8 μM in 100 μl) alone and the effects of GsMTx-4 when injected after SAR7334 (51 ± 8 μM in 100 μl). Abbreviations: BPi, blood pressure index calculated as the integrated blood pressure response to contraction; ∗*P* < 0.05 between the corresponding data points.

stretch in a rat model of peripheral artery disease than it had on this response in healthy rats. This was interpreted as Piezo 2 channels playing a greater role in rats with peripheral artery disease than in healthy rats. Moreover, Grotle et al. (2019) found that GsMTx-4 normalized the exaggerated pressor response to static stretch in a rat model of Type I diabetes. The authors also interpreted these findings as evidence supporting an exaggerated role

played by Piezo channels in evoking the exercise pressor reflex. While the present findings do not invalidate the findings of the above-mentioned studies, they warrant further investigation of the role played by TRPC6 in evoking the mechano- and exercise pressor reflex in these disease models; at the least they warrant reconsideration of the importance of the role played by Piezo 2 in evoking

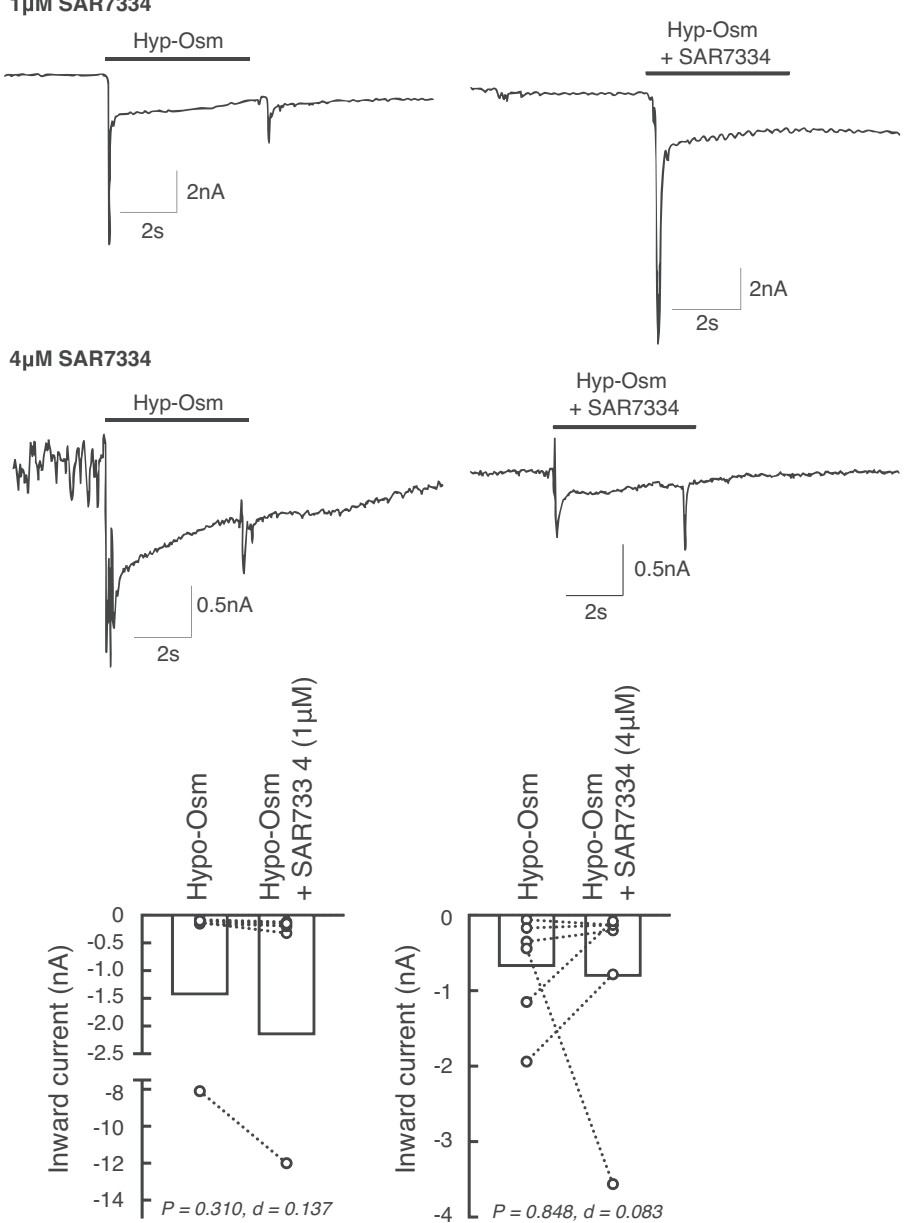

**Figure 6. Representative traces and individual effects of 1 μM and 4 μM of SAR7334 on the inward current of heterologous cells overexpressing Piezo 2 in response to membrane stretch**
Data are presented as individual (open dots) and group means (open bars) for the peak inward current evoked by membrane stretch. Membrane stretch was evoked by exposing the cells to hypoosmotic solution (Hypo-Osm). A more negative value indicates higher inward current. Two concentrations of SAR7334 were used, 1 μM and 4 μM, which corresponded to the calculated concentration of drugs reached in the interstium of myocytes following intra-arterial injections.

the exercise pressor reflex and the muscle mechanoreflex during static manoeuvres.

In our experiments, we investigated the overlapping effects of SAR7334 and GsMTx-4 by reversing the order of the drug injected (i.e. GsMTx-4 first and SAR7334 second). Consistent with our former experiments, we found that the effects of SAR7334 when injected secondary to GsMTx-4 were drastically reduced compared with its effects when injected alone (Fig. 5). These results are expected if both drugs shared a common effect on TRPC6 channels. However, the effects of SAR7334 were not completely abolished as SAR7334 further reduced the blood pressure response to static contraction when injected after GsMTx-4 (Fig. 2 right panel). This result suggests that the inhibition of TRPC6 by GsMTx-4 was submaximal, a possible consequence of the lesser potency and/or different mechanism of action of GsMTx-4 to inhibit TRPC6 channels compared with SAR7334. The GsMTx-4 IC$_{50}$ for TRPC6 is currently unknown (Spassova et al., 2006) whereas SAR7334 inhibit TRPC6 with an IC$_{50}$ of 7.9 nm (Maier et al., 2015). GsMTx-4, however, appears to act on TRPC6 by being inserted between the channel and the cell membrane to maintain TRPC6 in a closed state despite receptor binding or membrane stretch (Spassova et al., 2006). On the other hand, SAR7334 appears to close the pore of the TRPC6 channel to inhibit the flow of ions and cell depolarization (Rubaiy, 2024).

### SAR7334 does not consistently block heterologously expressed Piezo 2 channels

Our interpretation of the effects of GsMTx-4 and SAR7334 is valid only if SAR7334 did not inhibit Piezo 2 channels. Consequently, our patch-clamp experiments provided key findings that helped in the interpretation of our *in vivo* data. In the presence of 1 μm of SAR7334, the current evoked by membrane stretch in Piezo 2-expressing CHO cells was increased in the six cells tested without reaching statistical significance. In addition, in the presence of 4μm of SAR7334, the evoked current was increased in two cells and decreased in three cells. The overall change again did not reach statistical significance. SAR7334 therefore appeared to not inhibit Piezo 2 at 1 μm and to have a dual effect at 4 μm. Importantly, the concentrations of SAR7334 used during these patch-clamp experiments (1 and 4 μm) closely mimicked the concentrations of the drug used during our *in vivo* experiments for a rat weighing between ∼200 g and 500 g, respectively. Specifically, we calculated that the 7 μg/kg dose of SAR7334 injected in the superficial epigastric artery (∼80 μm for a 500 g rat) was diluted in 20 ml of volume which corresponded to the volume of the leg. The average weight of the animals and concentrations

of SAR7334 in the leg were 317 ± 47 g and 2.5 ± 0.4 μm, respectively. This calculation was performed instead of direct measurements because we could not effectively measure the concentration of SAR7334 reached in the interstitial space. This would have required microdialysis probes perfused with radiolabelled SAR7334 to calculate the precise recovery rate of the probes and antibodies to quantify the precise concentration of SAR7334 recovered; neither of the two are currently available. The lack of direct measure of SAR7334 concentration in the interstitial space represents a limitation of our protocol and should be kept in mind while interpreting our findings. Further experiments aiming at quantifying the dynamic changes of a compound concentration in muscles interstitial space when injected in the arterial circulation would be highly valuable.

Like GsMTx-4, which inhibits several mechano-sensitive channels, including TRPC6 (Spassova et al., 2006), SAR7334 could potentially have off-target effects. This raised the possibility that the effects of the drug were the consequences of the inhibition of channels other than TRPC6. If we cannot completely rule out this hypothesis for channels other than Piezo 2, we can confidently conclude that the effects of SAR7334 were not the results of blocking Piezo 2 channels at 1 μm. These results strengthen the findings of the present and previously published experiments that used SAR7334 and concluded that TRPC6 played an important role in evoking the pressor and sympathetic responses to the mechanical component of the exercise pressor reflex (Ducrocq et al., 2025).

### Methodological consideration

TRPC6 channels are expressed in smooth muscle cells and thus can play a role in regulating peripheral vaso-constriction/vasodilatation (Zulian et al., 2010). The reduced exercise pressor reflex or mechanoreflex found in response to SAR7334 could therefore be the consequence of impaired arterial vasoconstriction. However, our data showing that no difference in popliteal artery vascular conductance was observed between before and after TRPC6 inhibition suggest that the vasoconstriction was not altered. In addition, renal sympathetic nerve activity was inhibited by the drug showing that the reflexes were effectively inhibited.

Our *in vitro* experiments investigated the role of SAR7334 on Piezo 2 channels by stretching the cell membrane with hypoosmotic solution (Spassova et al., 2006). While this stimulus is effective in activating Piezo 2 channels (Fig. 6), it is rather distant from the mechanical stimulus of a contracting muscle which increases intra-muscular pressure as well as compresses rather than stretches afferent nerve endings (Crenshaw et al., 1995). This raises the possibility that different results could have

been obtained using a different stimulus that is more representative of muscle contraction.

While we assumed that the effects of GsMTx-4 and SAR7334 were the results of blocking TRPC6, we cannot rule out the hypothesis that both drugs shared another, common, off-target effect that would explain our results. For example, both drugs could act on TRPV4, another channel implicated in the mechanoreflex (Fukazawa et al., 2023). So far, no evidence supports this hypothesis. In addition, SAR7334 and GsMTx-4 could act on TRPC3, a channel that is closely related to TRPC6 (Maier et al., 2015). SAR7334 has been reported to block TRPC3 at concentration more than 20 times greater than TRPC6 (Maier et al., 2015); previously it has been suggested that TRPC3 could be activated by direct mechanical stimulation of cardiomyocytes (Yamaguchi et al., 2017). Although TRPC6 is expressed in dorsal root ganglia innervating the triceps surae muscles (Ducrocq et al., 2025), it remains unknown whether TRPC3 is also expressed and possesses similar properties in these tissues. Previous experiments found large expressions of TRPC3 in whole lumbar dorsal root ganglia, that were colocalized with IB-4 but not with Nf200 (Elg et al., 2007). The lack of colocalization of TRPC3 with Nf200, a marker of myelination (Fornaro et al., 2008; Ma, 2002), questions the mechanical transduction role played by TRPC3 in dorsal root ganglia innervating the skeletal muscles. Importantly, our present findings with SAR7334 closely replicate those reported previously with B1749327, another TRPC6 antagonist (Ducrocq et al., 2025). BI749327, which is structurally different from SAR7334, is 85-fold more selective for TRPC6 than it is for TRPC3 (Lin et al., 2019). In addition, the response to contraction or stretch was potentiated when a positive modulator was used (C20). C20 does not modulate TRPC3 or TRPC7 (Häfner et al., 2019). We consequently are confident that TRPC6 plays a role in evoking the mechanoreflex and that the probability that our results were due to TRPC3 inhibition was low. Although it is impossible to test every possible channel, these unresolved questions warrant further investigation into the mechanisms responsible for evoking both the exercise pressor reflex and the muscle mechanoreflex. This could be achieved using an alternative approach to pharmacological blockade, such as genetic knock out of Piezo 2 or TRPC6.

It could be suggested that differences in clearance rates between SAR7334 and GsMTx-4 could have explained the results of the present experiments. A previous study showed that the concentration of SAR7334 remained elevated in blood plasma for hours after administration (Maier et al., 2015). It appears unlikely that its effects decreased during the duration of our protocol, having a post-injection period of 20 min. Our data showing that the effects of SAR7334 was not different 20 min after the injection compared with 10 min after the injection

support that hypothesis (Ducrocq et al., 2025). We did not find data on the clearance of GsMTx-4 in tissue or blood plasma. Previous data showed that the concentration of the enantiomer of GsMTx-4 (GsMTx-4-D) in skeletal muscle remained elevated for days after a period of subcutaneous loading despite rapid blood clearance (Ward et al., 2018). In addition, previous experiments on the exercise pressor reflex that used GsMTx-4 and evoked contractions or stretches 30 min after the intra-arterial injection showed similar reductions as the ones reported in this manuscript 10 or 20 min after the injection (e.g. Copp et al., 2016a; Grotle et al., 2021). Consequently, it appears unlikely that differences in pharmacokinetics explained our results.

Sex difference in response to a physiological manoeuvre or treatment is an important topic in exercise physiology (D'Souza et al., 2023; Koba et al., 2012; Lee et al., 2023; Smith et al., 2019). In line with previous findings, we found that female rats displayed a reduced peak pressor response to both static contraction and passive stretch compared with that displayed by male rats (Ettinger et al., 1996; Smith et al., 2019). In addition, our previous study included only male rats (Ducrocq et al., 2025) and more data from female rats were needed on the role played by TRPC6 and GsMTx-4 (Grotle & Stone, 2025; Grotle et al., 2021). While this was not the primary aim of the present paper, we did not report any sex difference in the response to GsMTx-4 or SAR7334, suggesting that the role played by TRPC6 is not affected by the sex of the animal. Although the present experiment included at the minimum an equal number of female rats in the studied sample size, and we ensured that females were investigated during the diestrus phase of the estrous cycle (i.e. the phase with the lowest concentration of female-specific sex hormones) (Koba et al., 2012), our statistical analysis was likely underpowered and a greater sample size could have shed light on a possible difference.

## Conclusion

We found that blocking TRPC6 before injecting GsMTx-4, a stretch-activated mechanochannel antagonist, dramatically reduced the effects of GsMTx-4 when injected alone. In addition, our data show that the TRPC6 channel antagonist SAR7334 (1 μm) did not block the stretch-activated Piezo 2 currents in CHO cells, but rather increased the peak current. On the other hand, 4 μm SAR7334 exhibited a dual effect on the currents. Our results reinforced the already documented role played by TRPC6 channels in evoking the exercise pressor reflex and the muscle mechanoreflex. In addition, they suggest that the previously reported effects of GsMTx-4 during static manoeuvres may have involved TRPC6 and/or TRPC3 inhibition as well as Piezo 2 inhibition.

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

## Additional information

### Data availability statement

Authors state that data are available upon reasonable request.

### Conflict of interest

The authors declare no conflict of interest.

### Author contributions

G.P.D. designed the experiments. K.B. acquired the patch-clamp data. G.P.D. acquired, analysed the data and drafted the manuscript. All authors interpreted the data and revised the manuscript critically. All authors approved the final version of the manuscript and agree to be accountable for all aspects of the work in ensuring that questions related to the accuracy or integrity of any part of the work are appropriately investigated and resolved. All persons designated as authors qualify for authorship, and all those who qualify for authorship are listed.

### Fundings

This project is supported by the NIH (grant numbers R01HL161160, R01HL156594 and R01HL156513).

### Keywords

blood pressure, exercise mechanoreflex, exercise pressor reflex, Piezo 2, sympathetic nervous system, TRPC6

## Supporting information

Additional supporting information can be found online in the Supporting Information section at the end of the HTML view of the article. Supporting information files available:

**Peer Review History**

