## [Peer Review History · The Journal of Physiology]

GsMTx-4 inhibits the exercise pressor reflex and the muscle mechanoreflex primarily through TRPC inhibition

Guillaume P. Ducrocq, Laura Anselmi, Kristen Brandt, Jianli Wang, Victor Ruiz-Velasco, and Marc P Kaufman
DOI: 10.1113/JP289092

Corresponding author(s): Guillaume Ducrocq (g.ducrocq@live.fr)

Review Timeline:

Submission Date:	15-Apr-2025
Editorial Decision:	03-Jun-2025
Revision Received:	14-Jun-2025
Editorial Decision:	01-Jul-2025
Revision Received:	09-Jul-2025
Editorial Decision:	22-Jul-2025
Revision Received:	22-Jul-2025
Accepted:	31-Jul-2025

Senior Editor: Vaughan Macefield

Reviewing Editor: Mathew Piasecki

Transaction Report:

Dear Dr Ducrocq,

Re: JP-RP-2025-289092 "GsMTx-4 inhibit the exercise pressor reflex and the mechanoreflex primarily through TRPC6inhibition" by Guillaume P. Ducrocq, Laura Anselmi, Christine Brandt, Jian-Li Wang, Victor Ruiz-Velasco, and Marc P Kaufman

Thank you for submitting your manuscript to The Journal of Physiology. It has been assessed by a Reviewing Editor and by 2 expert referees and we are pleased to tell you that it is potentially acceptable for publication following satisfactory major revision.

LANGUAGE EDITING AND SUPPORT FOR PUBLICATION: If you would like help with English language editing, or other article preparation support, Wiley Editing Services offers expert help, including English Language Editing, as well as translation, manuscript formatting, and figure formatting at www.wileyauthors.com/eoo/preparation. You can also find resources for Preparing Your Article for general guidance about writing and preparing your manuscript at www.wileyauthors.com/eoo/prepresources.

REVISION CHECKLIST:

We look forward to receiving your revised submission.

Yours sincerely,

Vaughan Macefield
Senior Editor
The Journal of Physiology

REQUIRED ITEMS

- Author photo and profile. First or joint first authors are asked to provide a short biography (no more than 100 words for one author or 150 words in total for joint first authors) and a portrait photograph. These should be uploaded and clearly labelled together in a Word document with the revised version of the manuscript. See Information for Authors for further details.

- Your manuscript must include a complete Additional Information section, including competing interests; funding; author contributions and acknowledgements.

- Please ensure that the Article File you upload is a Word file.

- Please include an Abstract Figure file, as well as the Figure Legend text within the main article file. The Abstract Figure is a piece of artwork designed to give readers an immediate understanding of the research and should summarise the main conclusions. If possible, the image should be easily 'readable' from left to right or top to bottom. It should show the physiological relevance of the manuscript so readers can assess the importance and content of its findings. Abstract Figures should not merely recapitulate other figures in the manuscript. Please try to keep the diagram as simple as possible and without superfluous information that may distract from the main conclusion(s). Abstract Figures must be provided by authors no later than the revised manuscript stage and should be uploaded as a separate file during online submission labelled as File Type 'Abstract Figure'. Please also ensure that you include the figure legend in the main article file. All Abstract Figures should be created using BioRender. Authors should use The Journal's premium BioRender account to export high-resolution images. Details on how to use and access the premium account are included as part of this email.

- Please include a full title page as part of your main article (Word) file, which should contain the following: title, authors, affiliations, corresponding author name and contact details, keywords, and running title.

- The corresponding author must provide an institutional email address (not a personal address) for their author account. We encourage ALL co-authors to also provide institutional email addresses. If this cannot be provided (as corresponding author), then a stamped letter must be provided from the institution which confirms their role and employment there (please upload this with the revised submission).

EDITOR COMMENTS

Reviewing Editor:

Your paper has been reviewed by 2 experts in this area. Although both recognise the importance of the work there are a number of major concerns that must be addressed before publication can be considered. Please fully justify the reasoning behind the lack of quantification of SAR7334 concentration, and consider how this may impact the strength of your interpretation. Please also include baseline data for all animals.

I must stress that these concerns are significant, and should be fully addressed with any resubmission.

Please also see 'Required Items' above.

Senior Editor:

Thank you for submitting your manuscript to The Journal of Physiology. I have now received comments from two independent reviewers and a reviewing editor, all experts in the field. While the reviewers see merit in your work there are major methodological issues that need to be addressed. I invite you to submit a fully revised manuscript with point-by-point responses to each comment.

REFeree COMMENTS

Referee #1:

Dear authors, I would like to commend you on this relevant, original, and well-conceived study exploring the mechanisms underlying the mechanoreflex. The combination of in vivo physiological experiments with in vitro recordings is a notable strength, and your integration of pharmacological tools to probe mechanistic contributions is impressive. That said, I have several concerns regarding aspects of the experimental design, data interpretation, and statistical rigor that I believe should be carefully addressed to strengthen the manuscript. I hope the comments provided will be constructive and support you in refining and improving this promising work. I look forward to seeing your revised version.

Major Comments

I appreciate the effort the authors have made in designing a pharmacological approach to isolate the contributions of different channels. However, I would appreciate some further clarification on several points. Specifically, I am not entirely clear on the rationale for using serial injections with identical protocols. Did the authors consider potential channel saturation after the first injection, as well as differences in pharmacokinetics and half-lives between GsMTx-4 (a peptide with a likely short systemic half-life) and SAR7334 (a small molecule with a longer half-life), when developing the experimental approach? In other words, could the relatively small differences observed between injection orders be influenced, at least in part, by these pharmacokinetic differences rather than reflecting purely mechanistic separation?

Upon reviewing the graphs, I also noticed that there appear to be broadly similar patterns across both serial injection protocols, with a trend suggesting that subsequent injection of GsMTx-4 may reduce the inhibitory effect of SAR7334. It is interesting to note that the pressor responses are generally robust across experiments but seem somewhat larger when SAR7334 was injected after GsMTx-4 (Fig. 2, right panel), and that the strongest inhibitory effects appear to occur in cases with the largest pressor responses. Given the relatively small sample size and perhaps underpowered analysis, could the authors comment on how confident they are in drawing firm mechanistic conclusions from these patterns? Furthermore, did the authors consider the sample size needed to address the serial pattern given that this requires a more advanced analysis involving multiple comparison testing (i.e., ANOVA)?

While I appreciate that the authors aimed to match the number of molecules by using equimolar and equal-volume injections, could they clarify why this approach was selected over the protocol used in Copp et al., 2016? Additionally, have the authors fully considered the potential off-target effects of GsMTx-4 at higher dosages, as well as the known effects of SAR7334 on TRPC3 and TRPC7 (Maier et al., 2015, Br J Pharmacol)?

Finally, I was unable to locate any data showing the baseline physiological parameters (e.g., baseline MAP) before and after the serial injections. Was this information intentionally omitted? Including these baseline values would help demonstrate that the injections themselves did not alter resting conditions and would strengthen the confidence in the reported reflex responses.

Minor Points

The manuscript title and framing suggest the work conclusively isolates TRPC6, but the pharmacological blocker used (SAR7334) also affects other TRPC isoforms. Clarifying this point in the framing and interpretation would improve accuracy.

Line 37: Spelling error.

Could the authors explain why static muscle contraction was selected to study the mechanoreflex, rather than intermittent muscle contraction, which may elicit a stronger mechanoreflex stimulus?

I am also curious if the authors believe that some of the variability in inhibition between animals could reflect physiological redundancy in the mechanoreflex, with varying contributions from TRPC6 and Piezo channels across individuals.

Line 207: Pancuronium can sometimes lower blood pressure. Could the authors clarify whether it was administered twice during the protocol (once before the stretch procedure and again as a verification step at the end)?

Line 222: Did the authors exclude any data based on the stated exclusion criteria? Please clarify.

Line 262: Could the authors explain why they selected the Kolmogorov-Smirnov test for normality, given that the Shapiro-Wilk test is generally recommended for small samples?

Line 265: Please clarify whether the reported t-tests were one-tailed or two-tailed.

Figures: Why is "popliteal" reported for conductance but not for blood flow?

Referee #2:

The authors concluded that GsMTx-4 in our in vivo experiments primarily antagonized TRPC6 channels rather than Piezo 2 channels. This conclusion was based on in vivo and in vitro experiments in a rat model.

My primary concern is the concentration of the drug/peptide used in the study. SAR7334 has been shown to inhibit Ca²⁺ influx through TRPC6 channels, with an IC₅₀ value of 9.5 nM. The in vitro experiments (Figure 6) used 1 and 4 μM concentrations of SAR7334. The in vivo study used 32-80 μM, which the authors argued would be diluted by the amount of blood in the animal. However, the actual concentration in the animal was not measured. The authors acknowledged using much higher concentrations than those of Copp et al. (2016a). (Line 360). Generally, a high drug concentration is associated with less specific effects. Off-target or non-specific effects might have accounted for some of the results reported.

The critical experiments reported in Figure 6 also show the importance of concentration. In that figure, the authors used 1 and 4 μM of SAR7334. As the authors noted, the 4 μM experiment showed inconsistent results from cell to cell. One cell had a very large increase in current and is an outlier. All other cells (6 data pairs) showed no change or a current reduction. If the outlier is removed, the p-value will likely be < 0.05 in favor of the current reduction. If SAR7334 can inhibit the Piezo 2, giving a Piezo 2 inhibitor sMTx-4 after SAR7334 would lead to no further effects. The experiments should be done using an SAR7334 concentration that does not affect Piezo 2.

Minor comments:

A schematic of the surgical preparation and experimental setup would be helpful.

All studies were done at room temperature. If done at body temperature, will the results differ?

The figure legend should include the concentration of the drugs/peptides used. Currently, only Figure 6 clearly shows the concentration.

The authors stated that "Because no effect of sex was found, male and female were pooled in one group." Actual comparisons should be presented in the Results to support this statement. As noted in the Discussion, sex difference is an important topic. That statement is stronger if more data points are available to rule out the sex differences in these observations.

END OF COMMENTS

Dear authors, I would like to commend you on this relevant, original, and well-conceived study exploring the mechanisms underlying the mechanoreflex. The combination of in vivo physiological experiments with in vitro recordings is a notable strength, and your integration of pharmacological tools to probe mechanistic contributions is impressive. That said, I have several concerns regarding aspects of the experimental design, data interpretation, and statistical rigor that I believe should be carefully addressed to strengthen the manuscript. I hope the comments provided will be constructive and support you in refining and improving this promising work. I look forward to seeing your revised version.

Major Comments

I appreciate the effort the authors have made in designing a pharmacological approach to isolate the contributions of different channels. However, I would appreciate some further clarification on several points. Specifically, I am not entirely clear on the rationale for using serial injections with identical protocols. Did the authors consider potential channel saturation after the first injection, as well as differences in pharmacokinetics and half-lives between GsMTx-4 (a peptide with a likely short systemic half-life) and SAR7334 (a small molecule with a longer half-life), when developing the experimental approach? In other words, could the relatively small differences observed between injection orders be influenced, at least in part, by these pharmacokinetic differences rather than reflecting purely mechanistic separation?

Upon reviewing the graphs, I also noticed that there appear to be broadly similar patterns across both serial injection protocols, with a trend suggesting that subsequent injection of GsMTx-4 may reduce the inhibitory effect of SAR7334. It is interesting to note that the pressor responses are generally robust across experiments but seem somewhat larger when SAR7334 was injected after GsMTx-4 (Fig. 2, right panel), and that the strongest inhibitory effects appear to occur in cases with the largest pressor responses. Given the relatively small sample size and perhaps underpowered analysis, could the authors comment on how confident they are in drawing firm mechanistic conclusions from these patterns? Furthermore, did the authors consider the sample size needed to address the serial pattern given that this requires a more advanced analysis involving multiple comparison testing (i.e., ANOVA)?

While I appreciate that the authors aimed to match the number of molecules by using equimolar and equal-volume injections, could they clarify why this approach was selected over the protocol used in Copp et al., 2016? Additionally, have the authors fully considered the potential off-target effects of GsMTx-4 at higher dosages, as well as the known effects of SAR7334 on TRPC3 and TRPC7 (Maier et al., 2015, Br J Pharmacol)?

Finally, I was unable to locate any data showing the baseline physiological parameters (e.g., baseline MAP) before and after the serial injections. Was this

information intentionally omitted? Including these baseline values would help demonstrate that the injections themselves did not alter resting conditions and would strengthen the confidence in the reported reflex responses.

Minor Points

The manuscript title and framing suggest the work conclusively isolates TRPC6, but the pharmacological blocker used (SAR7334) also affects other TRPC isoforms. Clarifying this point in the framing and interpretation would improve accuracy.

Line 37: Spelling error.

Could the authors explain why static muscle contraction was selected to study the mechanoreflex, rather than intermittent muscle contraction, which may elicit a stronger mechanoreflex stimulus?

I am also curious if the authors believe that some of the variability in inhibition between animals could reflect physiological redundancy in the mechanoreflex, with varying contributions from TRPC6 and Piezo channels across individuals.

Line 207: Pancuronium can sometimes lower blood pressure. Could the authors clarify whether it was administered twice during the protocol (once before the stretch procedure and again as a verification step at the end)?

Line 222: Did the authors exclude any data based on the stated exclusion criteria? Please clarify.

Line 262: Could the authors explain why they selected the Kolmogorov–Smirnov test for normality, given that the Shapiro–Wilk test is generally recommended for small samples?

Line 265: Please clarify whether the reported t-tests were one-tailed or two-tailed.

Figures: Why is “popliteal” reported for conductance but not for blood flow?

Responses to reviewers

REQUIRED ITEMS

- Author photo and profile. First or joint first authors are asked to provide a short biography (no more than 100 words for one author or 150 words in total for joint first authors) and a portrait photograph. These should be uploaded and clearly labelled together in a Word document with the revised version of the manuscript. See Information for Authors for further details.

Response: First author profile and photo has been added.

- Your manuscript must include a complete Additional Information section, including competing interests; funding; author contributions and acknowledgements.

Response: A complete additional information section has now been added.

- Please ensure that the Article File you upload is a Word file.

- Please include an Abstract Figure file, as well as the Figure Legend text within the main article file. The Abstract Figure is a piece of artwork designed to give readers an immediate understanding of the research and should summarise the main conclusions. If possible, the image should be easily 'readable' from left to right or top to bottom. It should show the physiological relevance of the manuscript so readers can assess the importance and content of its findings. Abstract Figures should not merely recapitulate other figures in the manuscript. Please try to keep the diagram as simple as possible and without superfluous information that may distract from the main conclusion(s). Abstract Figures must be provided by authors no later than the revised manuscript stage and should be uploaded as a separate file during online submission labelled as File Type 'Abstract Figure'. Please also ensure that you include the figure legend in the main article file. All Abstract Figures should be created using BioRender. Authors should use The Journal's premium BioRender account to export high-resolution images. Details on how to use and access the premium account are included as part of this email.

Response: A graphical abstract has been added.

- Please include a full title page as part of your main article (Word) file, which should contain the following: title, authors, affiliations, corresponding author name and contact details, keywords, and running title.

Response: The title page has been fully completed.

- The corresponding author must provide an institutional email address (not a personal address) for their author account. We encourage ALL co-authors to also provide institutional email addresses. If this cannot be provided (as corresponding author), then a stamped letter must be provided from the institution which confirms their role and employment there (please upload this with the revised submission).

Response: An institutional email is now provided from all co-authors and first author.

EDITOR COMMENTS

Reviewing Editor:

Your paper has been reviewed by 2 experts in this area. Although both recognise the importance of the work there are a number of major concerns that must be

addressed before publication can be considered. Please fully justify the reasoning behind the lack of quantification of SAR7334 concentration, and consider how this may impact the strength of your interpretation. Please also include baseline data for all animals.

I must stress that these concerns are significant, and should be fully addressed with any resubmission.

Response: We thank the reviewing editor for handling our manuscript. We addressed fully all of the concerns that were made on our experiments. Baseline data for all variables are now provided in the manuscript. We believe the comments provided by the reviewers were meaningful and helped to improve the manuscript.

Please also see 'Required Items' above.

Response: all these items have been completed.

Senior Editor:

Thank you for submitting your manuscript to The Journal of Physiology. I have now received comments from two independent reviewers and a reviewing editor, all experts in the field. While the reviewers see merit in your work there are major methodological issues that need to be addressed. I invite you to submit a fully revised manuscript with point-by-point responses to each comment.

Response: We thank the senior editor for handling our manuscript. We addressed all major concerns. We believe the comments provided by the reviewers were meaningful and helped to improve our manuscript.

REFEREE COMMENTS

Referee #1:

Dear authors, I would like to commend you on this relevant, original, and well-conceived study exploring the mechanisms underlying the mechanoreflex. The combination of in vivo physiological experiments with in vitro recordings is a notable strength, and your integration of pharmacological tools to probe mechanistic contributions is impressive. That said, I have several concerns regarding aspects of the experimental design, data interpretation, and statistical rigor that I believe should be carefully addressed to strengthen the manuscript. I hope the comments provided will be constructive and support you in refining and improving this promising work. I look forward to seeing your revised version.

Response to reviewer 1:

We would like to thank reviewer 1 for their insights and enthusiastic comments about the originality and novelty of our data and experimental design. We respond below in blue font to each major and minor comments. We believe these comments greatly improved the manuscript.

Major Comments

I appreciate the effort the authors have made in designing a pharmacological approach to isolate the contributions of different channels. However, I would appreciate some further clarification on several points. Specifically, I am not entirely clear on the rationale for using serial injections with identical protocols.

Response to reviewer 1:

The goal of our serial injections of SAR7334 and GsMTx-4 was to isolate the contribution of Piezo 2 and TRPC6 channels in evoking the exercise pressor reflex and the mechanoreflex. GsMTx-4 is known for blocking Piezo 2 channels but it also blocks TRPC6 channels. We believe that the latter effect is less well recognized than is the former effect. Recently, TRPC6 channel was shown to play a role in evoking the mechanoreflex (Ducrocq et al 2024). Therefore, by injecting SAR7334 first, we attempted to block TRPC6 before injecting GsMTx-4 and thus isolate the effect of GsMTx-4 on Piezo 2.

Ducrocq GP, Anselmi L, Stella SL, Copp SW, Ruiz-Velasco V & Kaufman MP (2024). Inhibition and potentiation of the exercise pressor reflex by pharmacological modulation of TRPC6 in male rats. *J Physiol*; DOI: 10.1113/jp286118.

Did the authors consider potential channel saturation after the first injection, as well as differences in pharmacokinetics and half-lives between GsMTx-4 (a peptide with a likely short systemic half-life) and SAR7334 (a small molecule with a longer half-life), when developing the experimental approach? In other words, could the relatively small differences observed between injection orders be influenced, at least in part, by these pharmacokinetic differences rather than reflecting purely mechanistic separation?

Response to reviewer 1:

These are two valid and interesting comments. Regarding potential channel saturation, our protocol design aimed to diminish the role played by TRPC6 in evoking the mechanoreflex. To accomplish this, we injected SAR7334 first to isolate the role played by Piezo 2 when GsMTx-4 was injected. In that context, saturation of TRPC6 channel inhibition would have been ideal. However, this could not be verified in vivo. We chose the concentration of SAR7334 used in the present experiment based on our dose-response data showing that the inhibitory effects of SAR7334 starting to plateau without reaching mM concentrations which would have increased dramatically the risk of off target effects (Ducrocq et al 2024).

Regarding differences in pharmacokinetics, previous experiments showed that plasma concentrations of SAR7334 (Maier et al 2015) remained elevated for several hours after administration. It is unlikely that its effects decreased during the duration of our protocol having a post injection period of 20 minutes. Previous data from our laboratory showed that the effects of SAR7334 was not different 20 minutes after the injection compared to 10 minutes after injection (Ducrocq et al 2024). We did not find any data on the clearance of GsMTx-4 in tissue or blood plasma. Previous data showed that the concentration of the enantiomer of GsMTx-4 (GsMTx-4-D) in skeletal muscle remained elevated for days after a period of subcutaneous loading despite rapid blood clearance (Ward et al 2018). In addition, previous experiments on the exercise pressor reflex that used GsMTx-4 showed similar reduction in the mechanoreflex 30 minutes after the intra-arterial injection compared to our protocol (e.g. Copp et al. 2016; Grotle et al. 2021). Consequently, it is unlikely that the possible differences in pharmacokinetics explained our results.

We believe this latter comment to be highly relevant and added a paragraph in the methodological consideration section to discuss it. Specifically, it reads (L540-5554) : *“It could be suggested that differences in clearance rates between SAR7334 and GsMTx-4 could have explained the results of the present experiments. A previous study showed that the concentration of SAR7334 remained elevated in*

blood plasma for hours after administration (Maier et al., 2015). It appears unlikely that its effects decreased during the duration of our protocol having a post injection period of 20 minutes. Our data showing that the effects of SAR7334 was not different 20 minutes after the injection compared to 10 minutes after the injection support that hypothesis (Ducrocq et al., 2024). We did not find data on the clearance of GsMTx-4 in tissue or blood plasma. Previous data showed that the concentration of the enantiomer of GsMTx-4 (GsMTx-4-D) in skeletal muscle remained elevated for days after a period of subcutaneous loading despite rapid blood clearance (Ward et al., 2018). In addition, previous experiments on the exercise pressor reflex that used GsMTx-4 and evoked contractions or stretches 30 minutes after the intra-arterial injection showed similar reductions as the ones reported in this manuscript 10 or 20 minutes after the injection (e.g. Copp et al., 2016a; Grotle et al., 2021) . Consequently, it appears unlikely that differences in pharmacokinetics explained our results.”

Copp SW, Kim JS, Ruiz-Velasco V & Kaufman MP (2016). The mechano-gated channel inhibitor GsMTx4 reduces the exercise pressor reflex in decerebrate rats. *The Journal of physiology* **594**, 641–655.

Ducrocq GP, Anselmi L, Stella SL, Copp SW, Ruiz-Velasco V & Kaufman MP (2024). Inhibition and potentiation of the exercise pressor reflex by pharmacological modulation of TRPC6 in male rats. *J Physiol*; DOI: 10.1113/jp286118.

Grotle A-K, Huo Y, Harrison ML, Ybarbo KM & Stone AJ (2021). GsMTx-4 normalizes the exercise pressor reflex evoked by intermittent muscle contraction in early stage type 1 diabetic rats. *American journal of physiology Heart and circulatory physiology* **320**, H1738–H1748.

Maier T, Follmann M, Hessler G, Kleemann H-W, Hachtel S, Fuchs B, Weissmann N, Linz W, Schmidt T, Löhn M, Schroeter K, Wang L, Rütten H & Strübing C (2015). Discovery and pharmacological characterization of a novel potent inhibitor of diacylglycerol-sensitive TRPC cation channels. *British journal of Pharmacology* **172**, 3650–3660.

Ward CW, Sachs F, Bush ED & Suchyna TM (2018). GsMTx4-D provides protection to the D2.mdx mouse. *Neuromuscul Disord* **28**, 868–877.

Upon reviewing the graphs, I also noticed that there appear to be broadly similar patterns across both serial injection protocols, with a trend suggesting that subsequent injection of GsMTx-4 may reduce the inhibitory effect of SAR7334. Response to reviewer 1: We may have missed something, but our results showed that GsMTx-4 indeed significantly reduced the inhibitory effects of SAR7334. It can be seen in Fig. 5 that when SAR7334 was injected after GsMTx-4, its inhibitory effect was significantly reduced. We interpret this finding to mean that GsMTx-4 blocked TRPC6 channels. If GsMTx4 had no effect on TRPC6 channels, then the attenuation induced by SAR7334 should not have been present.

It is interesting to note that the pressor responses are generally robust across experiments but seem somewhat larger when SAR7334 was injected after GsMTx-4

(Fig. 2, right panel), and that the strongest inhibitory effects appear to occur in cases with the largest pressor responses.

Response to reviewer 1: The fact that the greatest inhibitory response was observed when the greatest pressor response was found could be due to 1) the fact that the greatest pressor responses were generally evoked by stronger contractions and thus greater mechanical stimulus and activation of TRPC6 channels; 2) a greater number of TRPC6 channels were available in a given animal, producing a greater pressor response and greater inhibition when the antagonists were injected; and 3) a statistical artifact. Regarding the latter, since our data are presented in absolute units and not in percent change, the difference in pre vs post drug injection has more of a chance to be high when the pressor response pre injection is high compared to a low pressor response. Notably, we also reported our drug effects in percents in the result section (i.e. relative changes).

Given the relatively small sample size and perhaps underpowered analysis, could the authors comment on how confident they are in drawing firm mechanistic conclusions from these patterns?

Response to reviewer 1: It is true that our sample size can be viewed as small, which is the result of minimizing animal use and the large effect of the drugs used. However, we feel very confident that our results are not due to low sample size, and/or underpowered analysis and that mechanistic conclusions can be drawn from our data.

- 1) Using the effect size reported in our previous experiment with SAR7334 (Ducrocq et al 2024; partial $\eta^2 = 0.526$ for contractions and partial $\eta^2 = 0.494$ for stretch experiments), a priori analysis of the required sample size estimated a number of 7 animals to detect a similar effect. This estimation can be easily recalculated using the free software G*power. The following parameters were used: F tests; ANOVA with repeated measures within factors; Group = 1; measures = 3; Correlation among rep = 0.01; $\alpha = 0.05$ and Power = 0.95).
- 2) We found statistical differences between our pre and post treatment responses. This suggest that our statistical power was de facto enough to detect the expected difference.

Ducrocq GP, Anselmi L, Stella SL, Copp SW, Ruiz-Velasco V & Kaufman MP (2024). Inhibition and potentiation of the exercise pressor reflex by pharmacological modulation of TRPC6 in male rats. *J Physiol*; DOI: 10.1113/jp286118.

Overall, the issue with low sample size comes when no statistical difference is found, which was not the case in most of our results.

Furthermore, did the authors consider the sample size needed to address the serial pattern given that this requires a more advanced analysis involving multiple comparison testing (i.e., ANOVA)?

Response to reviewer 1:

Yes we did consider the sample size required. The calculation used to estimate the required number of animals is detailed in our responses above. The manuscript has been updated to present our calculation (L 130-134). It is true that the ANOVA with repeated measures analysis is more advanced than standard paired t-tests, but this

sample size (between 6 and 12) is regularly used in the literature to effectively detect mechanistic effects (e.g. Anselmi et al. 2023; Butenas et al. 2022; 2024; Ducrocq et al. 2024; Fukazawa et al. 2023; Grotle et al. 2021).

Anselmi L, Ducrocq GP, Ruiz-Velasco V, Stocker SD, Higgins SP & Kaufman MP (2023). Functional knockout of the TRPV1 channel has no effect on the exercise pressor reflex in rats. *J Physiol* **601**, 5241–5256.

Butenas ALE, Parr SK, Flax JS, Carroll RJ, Baranczuk AM, Ade CJ, Hageman KS, Musch TI & Copp SW (2024). Protein kinase C epsilon contributes to chronic mechanoreflex sensitization in rats with heart failure. *J Physiol*; DOI: 10.1113/jp287020.

Butenas ALE, Rollins KS, Parr SK, Hammond ST, Ade CJ, Hageman KS, Musch TI & Copp SW (2022). Novel mechanosensory role for acid sensing ion channel subtype 1a in evoking the exercise pressor reflex in rats with heart failure. *J Physiol* **600**, 2105–2125.

Ducrocq GP, Anselmi L, Stella SL, Copp SW, Ruiz-Velasco V & Kaufman MP (2024). Inhibition and potentiation of the exercise pressor reflex by pharmacological modulation of TRPC6 in male rats. *J Physiol*; DOI: 10.1113/jp286118.

Fukazawa A, Hori A, Hotta N, Katanosaka K, Estrada JA, Ishizawa R, Kim H, Iwamoto GA, Smith SA, Vongpatanasin W & Mizuno M (2023). Antagonism of TRPV4 channels partially reduces mechanotransduction in rat skeletal muscle afferents. *J Physiol* **601**, 1407–1424.

Grotle A-K, Huo Y, Harrison ML, Ybarbo KM & Stone AJ (2021). GsMTx-4 normalizes the exercise pressor reflex evoked by intermittent muscle contraction in early stage type 1 diabetic rats. *American journal of physiology Heart and circulatory physiology* **320**, H1738–H1748.

While I appreciate that the authors aimed to match the number of molecules by using equimolar and equal-volume injections, could they clarify why this approach was selected over the protocol used in Copp et al., 2016?

Response to reviewer 1:

We are not sure what specific approach was used by Copp that should have been applied in our experimental design. If reviewer 1 is referring to the difference in concentration of GsMTx-4 and timings of injection compared to Copp's, we chose to match the concentration of GsMTx-4 and timings of injection *with those used for SAR7334* and not the opposite. This choice was made because we already documented the effects of the used concentration and timings of injection of SAR7334 on blood pressure and blood flow. Importantly, even if different timings and greater concentration of GsMTx-4 were used, our data showing the effects of GsMTx-4 alone closely replicated the results found by Copp et al.

Additionally, have the authors fully considered the potential off-target effects of GsMTx-4 at higher dosages, as well as the known effects of SAR7334 on TRPC3 and TRPC7 (Maier et al., 2015, Br J Pharmacol)?

Response to reviewer 1:

We considered possible off target effects of our drugs. SAR7334 can block TRPC3 and TRPC7 at concentrations that are more than 20 times greater than the concentration needed to block TRPC6 (Maier et al., 2015). It is possible that blockade of these channels occurred when SAR7334 was injected. While we cannot exclude that TRPC3 and TRPC7 played a role in the mechanoreflex, there is no evidence that TRPC7 is directly activated by a mechanical stimulus. There is evidence that TRPC3 is indirectly involved in inducing mechanical hypersensitivity (Tobori et al 2025), but its function is involved downstream of PLC activation rather than in direct mechanical transduction. Other experiments suggest that TRPC3 possesses a mechanical sensitivity in cardiomyocytes (Yamaguchi et al. 2017). Nevertheless, it remains unknown if TRPC3 is expressed and possesses the same property in dorsal root ganglia innervating the skeletal muscles. Previous experiments found large expressions of TRPC3 in whole lumbar DRGs, that were colocalized with IB-4 but not NF200 (Elg et al. 2007). The lack of colocalization of TRPC3 with NF200, a marker of myelination, questions the mechanical transduction role played by TRPC3 in DRG innervating the skeletal muscles. Importantly, in our previous experiment (Ducrocq et al. 2024), our results found with SAR7334 were replicated when a structurally different TRPC6 antagonist was used (BI749327). BI749327 is 85-fold more selective for TRPC6 than for TRPC3 (Lin et al 2019). In addition, the response to contraction or stretch was potentiated when a positive modulator was used (C20). C20 does not modulate TRPC3 or TRPC7 (Hafner et al 2019). We consequently are confident that TRPC6 plays a role in evoking the mechanoreflex and that the probability that our results were due to TRPC3 inhibition was low. However, this should be verified experimentally.

Ultimately, the fact that SAR7334 blocked TRPC3 or not does not change the outcomes of the manuscript which is “the effects of GsMTx-4 on the exercise pressor reflex and the mechanoreflex are mainly not mediated through inhibition of Piezo2”. All pharmacological tools are limited by their off-target effects. As mentioned in the methodological section of the original manuscript (L474-475 of original manuscript), further experiments are needed using genetic tools to reinforce our data.

A new paragraph in the methodological consideration has been added to the manuscript to discuss this important topic.

Specifically, it reads (L514-539) “*While we assumed that the effects of GsMTx-4 and SAR7334 were the results of blocking TRPC6, we cannot rule out the hypothesis that both drugs shared another, common, off target effect that would explain our results. For example, both drugs could act on TRPV4, another channel implicated in the mechanoreflex (Fukazawa et al., 2023). So far, no evidence supports this hypothesis. In addition, SAR7334 and GsMTx-4 could act on TRPC3, a channel that is closely related to TRPC6 (Maier et al., 2015). SAR7334 has been reported to block TRPC3 at concentration more than 20 times greater than TRPC6 (Maier et al., 2015); Previously it has been suggested that TRPC3 could be activated by direct mechanical stimulation of cardiomyocytes (Yamaguchi et al., 2017). Although TRPC6 is expressed in dorsal root ganglia innervating the triceps surae muscles (Ducrocq et al., 2024), it remains unknown if TRPC3 is also expressed and possesses similar properties in these tissues. Previous experiments found large expressions of TRPC3 in whole lumbar DRGs, that were colocalized with IB-4 but not with Nf200 (Elg et al., 2007) . The lack of colocalization of TRPC3 with Nf200, a marker of myelination (Ma, 2002; Fornaro et al., 2008), questions the mechanical transduction role played by TRPC3 in dorsal root ganglia innervating the skeletal muscles. Importantly, in our previous experiment (Ducrocq et al., 2024), our results found with SAR7334 were*

closely replicated when a structurally different TRPC6 antagonist was used (BI749327). BI749327 is 85-fold more selective for TRPC6 than for TRPC3 (Lin et al., 2019). In addition, the response to contraction or stretch was potentiated when a positive modulator was used (C20). C20 does not modulate TRPC3 or TRPC7 (Häfner et al., 2019). We consequently are confident that TRPC6 plays a role in evoking the mechanoreflex and that the probability that our results were due to TRPC3 inhibition was low. Although it is impossible to test every possible channel, these unresolved questions warrant further investigation into the mechanisms responsible for evoking both the exercise pressor reflex and the muscle mechanoreflex. This could be achieved using an alternative approach to pharmacological blockade, such as genetic knock out of Piezo 2 or TRPC6.”

Ducrocq GP, Anselmi L, Stella SL, Copp SW, Ruiz-Velasco V & Kaufman MP (2024). Inhibition and potentiation of the exercise pressor reflex by pharmacological modulation of TRPC6 in male rats. *J Physiol*; DOI: 10.1113/jp286118.

Elg S, Marmigere F, Mattsson JP & Ernfors P (2007). Cellular subtype distribution and developmental regulation of TRPC channel members in the mouse dorsal root ganglion. *The Journal of Comparative Neurology* **503**, 35–46.

Häfner S, Urban N & Schaefer M (2019). Discovery and characterization of a positive allosteric modulator of transient receptor potential canonical 6 (TRPC6) channels. *Cell calcium* **78**, 26–34.

Lin BL et al. (2019). In vivo selective inhibition of TRPC6 by antagonist BI 749327 ameliorates fibrosis and dysfunction in cardiac and renal disease. *Proceedings of the National Academy of Sciences of the United States of America* **116**, 10156–10161.

Maier T, Follmann M, Hessler G, Kleemann H-W, Hachtel S, Fuchs B, Weissmann N, Linz W, Schmidt T, Löhn M, Schroeter K, Wang L, Rütten H & Strübing C (2015). Discovery and pharmacological characterization of a novel potent inhibitor of diacylglycerol-sensitive TRPC cation channels. *British journal of pharmacology* **172**, 3650–3660.

Tobori S, Tamada K, Uemura N, Sawada K, Kakae M, Nagayasu K, Nakagawa T, Mori Y, Kaneko S & Shirakawa H (2025). Spinal TRPC3 promotes neuropathic pain and coordinates phospholipase C–induced mechanical hypersensitivity. *Proc Natl Acad Sci* **122**, e2416828122.

Yamaguchi Y, Iribe G, Nishida M & Naruse K (2017). Role of TRPC3 and TRPC6 channels in the myocardial response to stretch: Linking physiology and pathophysiology. *Prog Biophys Mol Biol* **130**, 264–272.

Finally, I was unable to locate any data showing the baseline physiological parameters (e.g., baseline MAP) before and after the serial injections. Was this information intentionally omitted? Including these baseline values would help demonstrate that the injections themselves did not alter resting conditions and would strengthen the confidence in the reported reflex responses.

Response to reviewer 1:

We are very grateful to reviewer 1 for spotting that baseline values were missing. This was *our mistake, and it wasn't intentional*. A table has been added that contain all the relevant information (Table 1).

Minor Points

The manuscript title and framing suggest the work conclusively isolates TRPC6, but the pharmacological blocker used (SAR7334) also affects other TRPC isoforms. Clarifying this point in the framing and interpretation would improve accuracy.

Response to reviewer 1: The title has been edited.

It now reads: **“GsMTx-4 inhibits the exercise pressor reflex and the *muscle mechanoreflex primarily through TRPC inhibition*”**

Line 37: Spelling error.

Response to reviewer 1: The text has been edited. Thank you.

Could the authors explain why static muscle contraction was selected to study the mechanoreflex, rather than intermittent muscle contraction, which may elicit a stronger mechanoreflex stimulus?

Response to reviewer 1:

The major difference in the mechanical stimulus between continuous and intermittent contraction mostly lies in the fact that muscle tension varies greatly during intermittent contraction whereas it does not during continuous contraction. In terms of “strength” of the mechanical stimulus, continuous contraction provides a stronger stimulus because the time under tension is at least twice greater than 1hz intermittent contraction. Specifically, we chose continuous contraction instead of intermittent contraction because 1) The effects of SAR7334 were already documented which helped us verifying that our previous results were reproducible; 2) The effects of SAR7334 on sympathetic activity response to continuous contraction was not measured previously; 3) The effects of SAR7334 are undocumented so far on intermittent contraction and it was unknown whether varying muscle tension implicated TRPC6 channels. Specific experiments should be conducted using intermittent contraction to extrapolate our findings to this type *of contraction*.

I am also curious if the authors believe that some of the variability in inhibition between animals could reflect physiological redundancy in the mechanoreflex, with varying contributions from TRPC6 and Piezo channels across individuals.

Response to reviewer 1:

We did not explore this hypothesis. We believe that several *factors* could explain the variability of inhibition between *animals*; *varying* expressions of TRPC6 and Piezo channels is one of them. The others could be that the surgery, the tension of the contraction or stretch, and/or the amount of injectate that actually *accessed* the triceps surae muscle differed between animals. Variability of the response to a treatment for a given animal is an interesting topic but is out of the scope of the present paper. With reviewer 1's approval we would prefer not to elaborate about it in the manuscript.

Line 207: Pancuronium can sometimes lower blood pressure. Could the authors clarify whether it was administered twice during the protocol (once before the stretch procedure and again as a verification step at the end)?

Response to reviewer 1:

The verification step that used pancuronium was not conducted during the stretch experiment because no stimulation of the tibial nerve was performed. This procedure was conducted for the contraction experiments only. One dose of pancuronium was injected before the stretch protocol. We updated the methods section to reflect that pancuronium was only used during the contraction experiments. Specifically, it reads (L232-234): "To show that tibial nerve stimulation did not electrically activate the axons of the group III and IV afferent fibers during the contraction experiments, we paralyzed the rat with pancuronium bromide (1mg/mL, 200 μ L; iv)."

Line 222: Did the authors exclude any data based on the stated exclusion criteria? Please clarify.

Response to reviewer 1: Yes. Two data points were excluded. It is now stated in the text (L243-244)

Line 262: Could the authors explain why they selected the Kolmogorov-Smirnov test for normality, given that the Shapiro-Wilk test is generally recommended for small samples?

Response to reviewer 1: We routinely used the Kolmogorov-Smirnov test for normality in the past and did not have this option on our statistical software (Statistica 8). We reanalyzed the samples using a more recent software (GraphPad Prism) and found generally similar results for normality. Several exceptions for which a non-parametric test needed to be used (Mann-Whitney U or Friedman tests) were found but the outcomes were similar if not more significant. The statistical analysis section has been updated to reflect these changes (L283-301).

Line 265: Please clarify whether the reported t-tests were one-tailed or two-tailed.

Response to reviewer 1: We used two tailed t-test. The text has been updated (L...)

Figures: Why is "popliteal" reported for conductance but not for blood flow?

Response to reviewer 1: There is no reason, it was just omitted. The legend has been updated. Thank you.

Referee #2:

The authors concluded that GsMTx-4 in our in vivo experiments primarily antagonized TRPC6 channels rather than Piezo 2 channels. This conclusion was based on in vivo and in vitro experiments in a rat model.

Response to reviewer 2: We would like to thank reviewer 2 for their review of our paper. We respond below in blue font to each major and minor comments. We believe these comments greatly improved the manuscript.

My primary concern is the concentration of the drug/peptide used in the study. SAR7334 has been shown to inhibit Ca(2+) influx through TRPC6 channels, with an IC50 value of 9.5 nM. The in vitro experiments (Figure 6) used 1 and 4 μ M concentrations of SAR7334. The in vivo study used 32-80 μ M, which the authors argued would be diluted by the amount of blood in the animal. However, the actual concentration in the animal was not measured. The authors acknowledged using much higher concentrations than those of Copp et al.(2016a). (Line 360). Generally,

a high drug concentration is associated with less specific effects. Off-target or non-specific effects might have accounted for some of the results reported.

Response to reviewer 2: Off-target effects are an inherent limitation of pharmacological antagonists and should be carefully considered when interpreting data extracted from experiments. We cannot rule out that SAR7334 and/or GsMTx-4 inhibited channels other than their respective targets, namely TRPC6 and Piezo-2 channels. This topic is specifically discussed now in the revised manuscript (L514-539). However, even if SAR7334 or GsMTx-4 had off target effects, this does not change the conclusion of our experiments which is that the effects of SAR7334 removed almost entirely the effects of GsMTx-4. Both drugs appear to share a common target which evidence from previous experiments and our in vitro data suggest is not Piezo 2 channel, but is the TRPC6 channel (Spassova et al. 2006). It can always be stated that both drugs could have a different common target, but at the moment there is no evidence for that and we cannot rule out every possible channel. Overall, our results questioned the role attributed to Piezo 2 in playing a major role in evoking the exercise pressor reflex and the mechanoreflex.

Regarding the fact that the concentration of SAR7334 was not measured in vivo, this experiment is not feasible at the moment. We believe that the only way to obtain these data would be through collecting interstitial fluid from microdialysis probes inserted in the triceps surae muscles. To quantify the precise concentration of SAR7334, a radiolabeled SAR7334 or an ELISA assay kit should have been used. Neither of the two are currently available and experiment using radioactive compound is strongly discouraged in our institute. Not knowing the exact concentration of SAR7334 reached in the triceps surae muscles is a limit from our protocol that is now discussed in the methodological consideration section (L482-487).

Specifically it reads: “This calculation *was performed instead of direct measurements because we could not effectively measure the concentration of SAR7334 reached in the interstitial space. This would have required microdialysis probes perfused with radiolabeled SAR7334 to calculate the precise recovery rate of the probes and antibodies to quantify the precise concentration of SAR7334 recovered; neither of the two are currently available.*”

The critical experiments reported in Figure 6 also show the importance of concentration. In that figure, the authors used 1 and 4 μM of SAR7334. As the authors noted, the 4 μM experiment showed inconsistent results from cell to cell. One cell had a very large increase in current and is an outlier. All other cells (6 data pairs) showed no change or a current reduction. If the outlier is removed, the p-value will likely be < 0.05 in favor of the current reduction. If SAR7334 can inhibit the Piezo 2, giving a Piezo 2 inhibitor sMTx-4 after SAR7334 would lead to no further effects. The experiments should be done using an SAR7334 concentration that does not affect Piezo 2.

Response to reviewer 2: One cell indeed presented a large increase in current. As stated in the manuscript, 3 cells decreased, 1 cell did not change and 1 other cell increased its current among the remaining 5 cells.

Except for the fact that the current increase of that cell was large, we had no valid reason to remove this data point. Especially since an increase in current was consistent with the results of the experiments that used 1 μM concentrations during which several cells showed an increase in current following exposure to SAR7334. Regardless, even when this data point is removed, no statistical significance was found ($P = 0.1520$). Importantly, the 4 μM concentration represent the dose that would

be reached for a 500g rats. The heaviest rat that was used in our experiment weighted 430g representing an estimated concentration of 3.44 uM of SAR7334 in the leg. Considering this and the fact that statistical significance was not reached even when the cell that potentiated was removed from the analysis, it cannot be concluded that SAR7334 consistently blocked Piezo 2 channels.

Minor comments:

A schematic of the surgical preparation and experimental setup would be helpful.
Response to reviewer 2: A graphical abstract was submitted with the revision manuscript that present the experimental setup.

All studies were done at room temperature. If done at body temperature, will the results differ?

Response to reviewer 2: We haven't thought about the effect of room temperature on our results. However, we are not aware of any thermosensitivity from TRPC6 and Piezo channels. Consequently, changing room temperature shouldn't have much effects in the role of TRPC6 and Piezo channels.

The figure legend should include the concentration of the drugs/peptides used. Currently, only Figure 6 clearly shows the concentration.

Response to reviewer 2: The figure legends have been updated.

The authors stated that "Because no effect of sex was found, male and female were pooled in one group." Actual comparisons should be presented in the Results to support this statement. As noted in the Discussion, sex difference is an important topic. That statement is stronger if more data points are available to rule out the sex differences in these observations.

Response to reviewer 2: A specific section and table on sex difference has been added to the results section (L353-363). In addition, female data points are now presented in closed circles in the bar graph figures.

Dear Dr Ducrocq,

Re: JP-RP-2025-289092R1 "**GsMTx-4 inhibits the exercise pressor reflex and the muscle mechanoreflex primarily through TRPC inhibition**" by Guillaume P. Ducrocq, Laura Anselmi, Kristen Brandt, Jianli Wang, Victor Ruiz-Velasco, and Marc P Kaufman

Thank you for submitting your manuscript to The Journal of Physiology. It has been assessed by a Reviewing Editor and by 2 expert referees and we are pleased to tell you that it is potentially acceptable for publication following satisfactory major revision.

LANGUAGE EDITING AND SUPPORT FOR PUBLICATION: If you would like help with English language editing, or other article preparation support, Wiley Editing Services offers expert help, including English Language Editing, as well as translation, manuscript formatting, and figure formatting at www.wileyauthors.com/eoo/preparation. You can also find resources for Preparing Your Article for general guidance about writing and preparing your manuscript at www.wileyauthors.com/eoo/prepresources.

REVISION CHECKLIST:

We look forward to receiving your revised submission.

Yours sincerely,

Vaughan Macefield
Senior Editor
The Journal of Physiology

EDITOR COMMENTS

Reviewing Editor:

Comments for Authors to ensure the paper complies with the Statistics Policy (Required):
Please show individual data, rather than mean (SD) in Figures 1 and 3.

Comments to the Author:

Your revised manuscript has been evaluated by the original reviewers. Although many comments have been addressed, there remain a number of areas of concern.

Please demonstrate how your results support the notion that GsMTx-4 primarily antagonized TRPC6 channels rather than Piezo 2 channels, as highlighted by both reviewers.

I stress that the manuscript cannot be considered for publication until all concerns are fully addressed.

Senior Editor:

Comments for Authors to ensure the paper complies with the Statistics Policy (Required):
Data points need to be provided in Figs 1 and 3.

Comments to the Author:

Thank you for submitting your revised manuscript to The Journal of Physiology. As you will see from the comments of the two independent reviewers and the reviewing editor, some issues remain, particularly with respect to the pharmacological interventions and their specificity, given that available evidence suggests that GsMTx-4 has a much more modest inhibitory effect under static conditions; this should be more clearly acknowledged when interpreting the involvement of Piezo and TRPC channels.

REFEREE COMMENTS

Referee #1:

Thank you for the thoughtful revision. The manuscript is substantially improved, and I appreciate the authors' careful responses to my initial comments. That said, I have a few additional suggestions and clarifications for the current version that I encourage the authors to consider. I have arranged these in the order presented in the manuscript, not importance.

- The study included both sexes, yet the abstract refers only to male rats. Please revise for consistency and clarity.
- While baseline data are now included, the authors only report a single p-value, presumably the lowest observed, without showing individual comparisons. For transparency, I recommend adding a column with the individual p-values in the table. Also strange that this is inconsistent across tables.
- Some abbreviations and units are unclear in the tables. For instance, what does "Ø" represent? Is "G" referring to grams of tension? If so, the values seem unusually low (e.g., 80 g). Please check for accuracy and clarify units and abbreviations in all tables.
- While I agree that sex-based comparisons are valuable, the study appears underpowered for detecting such differences and was not specifically designed for that aim. I wonder if summarizing in text would be more appropriate than focusing on them in a table. Alternatively, the discussion tempered. If keeping the table, it is unclear why the table includes peak pressor responses only for the pre-blockade condition, but not post-blockade?
- Keywords: Please correct the spelling of "sympathetic."
- Mechanism of TRPC inhibition: One point remains unclear, TRPC inhibition following GsMTx-4 injection still reduces the pressor response to both contraction and stretch. If GsMTx-4 truly acts via TRPC antagonism, why is there an additional effect with SAR7334? Could this indicate that GsMTx-4 only partially blocks TRPC channels, or targets a different subset than SAR7334? Also, SAR7334 appears to have a markedly stronger inhibitory effect than GsMTx-4, any thoughts on this? Also, could this be relevant in regards to my last point below?
- More curiosity, the inhibitory effects seem weaker on RSNA compared to MAP, particularly during stretch. Do the authors have any interpretation or hypothesis for this observation?
- Mechanistic interpretation and Piezo channels: I'd like to revisit a key point from my previous review. The approach used to isolate TRPC and Piezo contributions may not be optimal, particularly given that GsMTx-4's inhibitory effect appears limited under static conditions (i.e., sustained stretch or contraction). While the authors briefly acknowledge this (lines 417-424), I believe the current discussion does not fully or accurately reflect the available evidence (line 415-424). For example, the manuscript frames Piezo channels as unlikely mediators of GsMTx-4 effects on the mechanoreflex. However, experiments in healthy animals such as Copp et al. (2015) show only transient inhibition during the initial 5 seconds of tendon stretch, while Grotle et al. (2019, PMID: 30840487, Fig. 3) found no inhibition at all under similar conditions. In contrast, stronger effects of GsMTx-4 are observed during dynamic stimuli (e.g., intermittent contraction or dynamic stretch) or in disease models. Taken together, I could speculate that this pattern suggests TRPC are more important during the static component of the reflex, while Piezo channels may contribute more to the dynamic component of the reflex, or under pathological conditions, but not to the static component under healthy conditions. I encourage the authors to consider the points in their interpretation of findings in light of available data.
- Please review line 438 and 442 for accuracy. Copp (2016) and Grotle (2019) used static stretch not intermittent stretch. Copp 2016 and Grotle 2021 used intermittent contraction.

Referee #2:

The authors stated, "It cannot be concluded that SAR7334 consistently blocked Piezo 2 channels." That statement acknowledges the possibility that SAR7334 at 4 μ M would inconsistently (i.e. erratically or unpredictably) block piezo 2. If piezo 2 is inconsistently blocked, adding GsMTx-4 would have inconsistent effects (Figure 4, left lower panel). These results cannot be used to support the conclusion that GsMTx-4 primarily antagonized TRPC6 channels rather than Piezo 2 channels.

END OF COMMENTS

EDITOR COMMENTS

Reviewing Editor:

Comments for Authors to ensure the paper complies with the Statistics Policy (Required):

Please show individual data, rather than mean (SD) in Figures 1 and 3.

Response to reviewing editor: Thank you for your comment. Figure 1 and 3 present the time courses of MAP, tension, blood flow and RSNA on 2s-by-2s time scale (17 data points). Providing individual data in every condition (pre, post 1st injection and post 2nd injection) mean that we would have to plot between 408 and 510 data points on each figure depending on the animal number (between 7 and 9). This would have a large detrimental impact on the readability of the figures and seems unrealistic. However, we would be happy to explore alternative options if necessary.

Comments to the Author:

Your revised manuscript has been evaluated by the original reviewers. Although many comments have been addressed, there remain a number of areas of concern.

Please demonstrate how your results support the notion that GsMTx-4 primarily antagonized TRPC6 channels rather than Piezo 2 channels, as highlighted by both reviewers. I stress that the manuscript cannot be considered for publication until all concerns are fully addressed.

Response: We thank the reviewing editor for handling our manuscript. We addressed the points raised by both reviewers. Our responses can be found in blue font below.

Senior Editor:

Comments for Authors to ensure the paper complies with the Statistics Policy (Required):

Data points need to be provided in Figs 1 and 3.

Response to senior editor: Thank you for your comment. Figure 1 and 3 present the time courses of MAP, tension, blood flow and RSNA on 2s-by-2s time scale (17 data points). Providing individual data in every condition (pre, post 1st injection and post 2nd injection) mean that we would have to plot between 408 and 510 data points on each figure depending on the animal number (between 7 and 9). This would have a large detrimental impact on the readability of the figures and seems unrealistic. However, we would be happy to explore alternative options to comply with the statistical policies.

Comments to the Author:

Thank you for submitting your revised manuscript to The Journal of Physiology. As you will see from the comments of the two independent reviewers and the reviewing editor, some issues remain, particularly with respect to the pharmacological interventions and their specificity, given that available evidence suggests that GsMTx-4 has a much more modest inhibitory effect under static conditions; this should be more clearly acknowledged when interpreting the involvement of Piezo and TRPC channels.

Response: We thank the senior editor for handling our manuscript. We addressed the points raised by both reviewers. Our responses can be found in blue font below.

REFEREE COMMENTS

Referee #1:

Thank you for the thoughtful revision. The manuscript is substantially improved, and I appreciate the authors' careful responses to my initial comments. That said, I have a few additional suggestions and clarifications for the current version that I encourage the authors to consider. I have arranged these in the order presented in the manuscript, not importance.

Response to reviewer 1: We thank reviewer 1 for their thoughtful review of our manuscript. A point-by-point response to their comments is provided below.

- The study included both sexes, yet the abstract refers only to male rats. Please revise for consistency and clarity.

Response to reviewer 1: The abstract has been edited.

- While baseline data are now included, the authors only report a single p-value, presumably the lowest observed, without showing individual comparisons. For transparency, I recommend adding a column with the individual p-values in the table. Also strange that this is inconsistent across tables.

Response to reviewer 1: We added the lowest P value for each index and experiments. We didn't have the space available to add each individual P values, but we believe that this format provides enough details. Nevertheless, we would be willing to revise this table if reviewer 1 insists.

- Some abbreviations and units are unclear in the tables. For instance, what does "Ø" represent? Is "G" referring to grams of tension? If so, the values seem unusually low (e.g., 80 g). Please check for accuracy and clarify units and abbreviations in all tables.

Response to reviewer 1: "Ø" was referring to the fact that no measure was made for blood flow during the stretch experiments (in table 1). "Ø" has been changed for "not measured". In table 2, "Ø" referred to the fact that "n" does not have units since it represents the count of animal per group. "Ø" has been changed for "none". "G" was a typo and was supposed to be "g" for grams. The text has been edited. The values presented are not unusual given that we wrote in the method section that baseline tension was set between 60 and 100g of tension. Baseline tension being equal to ~80g respect this guideline.

- While I agree that sex-based comparisons are valuable, the study appears underpowered for detecting such differences and was not specifically designed for that aim. I wonder if summarizing in text would be more appropriate than focusing on them in a table. Alternatively, the discussion tempered. If keeping the table, it is unclear why the table includes peak pressor responses only for the pre-blockade condition, but not post-blockade?

Response to reviewer 1: We agree that the statistical analyses on the role of sex differences on the effects of SAR7334 and GsMTx-4 was underpowered and decided to remove the part of the table that was related to them. However, we cannot consider that the comparison of males vs females before the first injection was underpowered given that we found a statistical difference and had twice the sample size. We therefore decided to keep this part of the table.

- Keywords: Please correct the spelling of "sympathetic."

Response to reviewer 1: We may have missed something, but we couldn't find any misspelling of this word.

- Mechanism of TRPC inhibition: One point remains unclear, TRPC inhibition following GsMTx-4 injection still reduces the pressor response to both contraction and stretch. If GsMTx-4 truly acts via TRPC antagonism, why is there an additional effect with SAR7334? Could this indicate that GsMTx-4 only partially blocks TRPC channels, or targets a different subset than SAR7334? Also, SAR7334 appears to have a markedly stronger inhibitory effect than GsMTx-4, any thoughts on this? Also, could this be relevant in regards to my last point below?

Response to reviewer 1: In line with Reviewer 1's comment, we indeed speculate that the additional effect of SAR7334 was due to the fact that GsMTx-4 is not as potent as SAR7334 and only partially block TRPC channels. Therefore, a further reduction was found when SAR7334 was injected after GsMTx-4. This greater potency to TRPC inhibition might also explain why SAR7334 have a larger effect than GsMTx-4. This was discussed already in the original manuscript (L453-467, now L454-468)

- More curiosity, the inhibitory effects seem weaker on RSNA compared to MAP, particularly during stretch. Do the authors have any interpretation or hypothesis for this observation?

Response to reviewer 1: We are unsure if a direct magnitude comparison between MAP and RSNA is feasible. RSNA is only one of many sympathetic outflows that are activated by the exercise pressor reflex. Moreover, RSNA is greatly impacted by the arterial baroreflex which is activated by the pressor response to muscle contraction.

- Mechanistic interpretation and Piezo channels: I'd like to revisit a key point from my previous review. The approach used to isolate TRPC and Piezo contributions may not be optimal, particularly given that GsMTx-4's inhibitory effect appears limited under static conditions (i.e., sustained stretch or contraction). While the authors briefly acknowledge this (lines 417-424), I believe the current discussion does not fully or accurately reflect the available evidence (line 415-424). For example, the manuscript frames Piezo channels as unlikely mediators of GsMTx-4 effects on the mechanoreflex. However, experiments in healthy animals such as Copp et al. (2015) show only transient inhibition during the initial 5 seconds of tendon stretch, while Grotle et al. (2019, PMID: 30840487, Fig. 3) found no inhibition at all under similar conditions. In contrast, stronger effects of GsMTx-4 are observed during dynamic stimuli (e.g., intermittent contraction or dynamic stretch) or in disease models. Taken together, I could speculate that this pattern suggests TRPC are more important during the static component of the reflex, while Piezo channels may contribute more to the dynamic component of the reflex, or under pathological conditions, but not to the static component under healthy conditions. I encourage the authors to consider the points in their interpretation of findings in light of available data.

Response to reviewer 1: Thank you for this comment. While the role played by Piezo channels in evoking the exercise pressor reflex and mechanoreflex during intermittent contraction or stretch is important to elucidate, this was not the purpose of the present experiment. We believe our protocol, although not perfect, was adequate to isolate the role played by piezo channels in evoking the exercise pressor reflex and mechanoreflex during static contraction or stretch. We edited our discussion to reflect that our data do not refute the findings concerning intermittent

contraction and stretch and are limited to static (sustained) contraction and stretch. The edited discussion is located L425-429 and Line 433-453.

- Please review line 438 and 442 for accuracy. Copp (2016) and Grotle (2019) used static stretch not intermittent stretch. Copp 2016 and Grotle 2021 used intermittent contraction.

Response to reviewer 1: You are correct, and we edited the text. Thank you for spotting that mistake.

Referee #2:

The authors stated, "It cannot be concluded that SAR7334 consistently blocked Piezo 2 channels." That statement acknowledges the possibility that SAR7334 at 4 uM would inconsistently (i.e. erratically or unpredictably) block piezo 2. If piezo 2 is inconsistently blocked, adding GsMTx-4 would have inconsistent effects (Figure 4, left lower panel). These results cannot be used to support the conclusion that GsMTx-4 primarily antagonized TRPC6 channels rather than Piezo 2 channels.

Response to reviewer 2: We thank reviewer 2 for their comment. We believe they already provided a response to their comment on the effect of SAR7334 being uncertain at 4uM. For the alternative statistical hypothesis that suggests SAR7334 blocks Piezo 2 channels, our P value ($P = 0.848$) was not low enough to accept it. We therefore accepted the hypothesis that the effect of SAR7334 is null on Piezo 2 channels. This is the approach adopted by the scientific community, and is the one we also endorsed. Our interpretation cannot be based on "what ifs". In addition, our interpretation was not solely based on our 4uM data but also on 1) our data showing that at 1uM SAR7334 did not block any of the cells tested, 2) that the calculated concentration of SAR7334 reached at the interstitial level never reached a concentration greater than 3.44uM with an averaged concentration of 2.5 ± 0.4 uM (~40% lower than 4uM in average), and 3) that contrary to what reviewer 2 suggests, our results on the effects of SAR7334 appear to be consistent during contraction or stretch between animals and experiments in vivo (Ducrocq et al 2024). Furthermore, the **inhibitory** effects of GsMTx-4 were exacerbated after SAR7334 was injected. If the effects of SAR7334 were erratic or unpredictable, these results could not have been found and the unpredictability would have been reflected on our in vivo data. It was not the case. Our approach was not perfect, and we acknowledged it in the manuscript. We have been very cautious in our interpretation and discussion of the data and highlighted that further, complementary, data were needed. For example, data from genetically modified animals.

Dear Dr Ducrocq,

Re: JP-RP-2025-289092R2 "**GsMTx-4 inhibits the exercise pressor reflex and the muscle mechanoreflex primarily through TRPC inhibition**" by Guillaume P. Ducrocq, Laura Anselmi, Kristen Brandt, Jianli Wang, Victor Ruiz-Velasco, and Marc P Kaufman

Thank you for submitting your manuscript to The Journal of Physiology. It has been assessed by a Reviewing Editor and by 2 expert referees and we are pleased to tell you that it is acceptable for publication following satisfactory minor revision.

REVISION CHECKLIST:

We look forward to receiving your revised submission.

Yours sincerely,

Vaughan Macefield
Senior Editor
The Journal of Physiology

EDITOR COMMENTS

Reviewing Editor:

Thank you for addressing all comments.

As highlighted by reviewer #2, please include the limitation of being unable to accurately determine in vivo concentrations.

Senior Editor:

Thank you for your detailed responses to the reviewers. Before we can proceed, please add commentary on the limitations, as requested by Reviewer 2.

REFEREE COMMENTS

Referee #1:

I thank the authors for addressing my comments and commend them on a well-executed study. I have no further comments.

Referee #2:

The authors argued that the 4 μM in vitro data may not apply to the in vivo experiment because they calculated the in vivo concentration to be around $2.5 \pm 0.4 \mu\text{M}$, which is lower than 4 μM . A limitation to this argument is that the in vivo concentration could not be accurately determined because the techniques are currently unavailable (page 23, last few lines). I suggest that the authors include a limitation section in the Discussion to highlight the various limitations of the study, including the inability to measure the concentrations in vivo.

END OF COMMENTS

EDITOR COMMENTS

Reviewing Editor:

Thank you for addressing all comments.

As highlighted by reviewer #2, please include the limitation of being unable to accurately determine in vivo concentrations.

Response to the reviewing editor: We thank the reviewing editor for accepting our manuscript and for his handling and review of our manuscript. Both referees provided constructive and important comments and suggestion that helped to improve the quality and depth of our manuscript. We addressed reviewer 2's suggestion. We did not include these sentences in the "methodological consideration" section of the discussion because we were already discussing the fact that we did not measure interstitial concentration of SAR7334. Specifically, it reads (added sentences in italic font; L484-492):

"This calculation was performed instead of direct measurements because we could not effectively measure the concentration of SAR7334 reached in the interstitial space. This would have required microdialysis probes perfused with radiolabeled SAR7334 to calculate the precise recovery rate of the probes and antibodies to quantify the precise concentration of SAR7334 recovered; neither of the two are currently available. *The lack of direct measure of SAR7334 concentration in the interstitial space represents a limitation of our protocol and should be kept in mind while interpreting our findings. Further experiments aiming at quantifying the dynamic changes of a compound concentration in muscles interstitial space when injected in the arterial circulation would be highly valuable.*"

Senior Editor:

Thank you for your detailed responses to the reviewers. Before we can proceed, please add commentary on the limitations, as requested by Reviewer 2.

Response to senior editor: We thank the senior editor for accepting our manuscript and for his handling and review of our manuscript. We believe the reviews from both referees were constructive and help dramatically improved the quality and depth of the manuscript.

REFEREE COMMENTS

Referee #1:

I thank the authors for addressing my comments and commend them on a well-executed study. I have no further comments.

Response to reviewer 1: We thank reviewer 1 of their suggestions and comments that helped dramatically improved the quality and depth of our manuscript.

Referee #2:

The authors argued that the 4 μM in vitro data may not apply to the in vivo experiment because they calculated the in vivo concentration to be around 2.5 ± 0.4 μM , which is lower than 4 μM . A limitation to this argument is that the in vivo concentration could not be accurately determined because the techniques are currently unavailable (page 23, last few lines). I suggest that the authors include a limitation section in the Discussion to highlight the various limitations of the study, including the inability to measure the concentrations in vivo.

Response to reviewer 2: Thank you for this, and previous, suggestions and comments. We believe reviewer 2's review helped dramatically improved our manuscript. We added a few sentences in the discussion to emphasize the fact that SAR7334 concentration was not measured. We did not include these sentences in the "methodological consideration" section of the discussion because we already discussed this matter in the previous section. Specifically, it reads (added sentences in italic font; L484-492):

"This calculation was performed instead of direct measurements because we could not effectively measure the concentration of SAR7334 reached in the interstitial space. This would have required microdialysis probes perfused with radiolabeled SAR7334 to calculate the precise recovery rate of the probes and antibodies to quantify the precise concentration of SAR7334 recovered; neither of the two are currently available. *The lack of direct measure of SAR7334 concentration in the interstitial space represents a limitation of our protocol and should be kept in mind while interpreting our findings. Further experiments aiming at quantifying the dynamic changes of a compound concentration in muscles interstitial space when injected in the arterial circulation would be highly valuable.*"

Dear Dr Ducrocq,

Re: JP-RP-2025-289092R3 "**GsMTx-4 inhibits the exercise pressor reflex and the muscle mechanoreflex primarily through TRPC inhibition**" by Guillaume P. Ducrocq, Laura Anselmi, Kristen Brandt, Jianli Wang, Victor Ruiz-Velasco, and Marc P Kaufman

We are pleased to tell you that your paper has been accepted for publication in The Journal of Physiology.

Yours sincerely,

Vaughan Macefield
Senior Editor
The Journal of Physiology

If you would like to receive our 'Research Roundup', a monthly newsletter highlighting the cutting-edge research published in The Physiological Society's family of journals (The Journal of Physiology, Experimental Physiology, Physiological Reports, The Journal of Nutritional Physiology and The Journal of Precision Medicine: Health and Disease), please click this link, fill in your name and email address and select 'Research Roundup':
<https://www.physoc.org/journals-and-media/membernews>

- You can help your research get the attention it deserves! Check out Wiley's free Promotion Guide for best-practice recommendations for promoting your work at: www.wileyauthors.com/eo/guide. You can learn more about Wiley Editing Services which offers professional video, design, and writing services to create shareable video abstracts, infographics, conference posters, lay summaries, and research news stories for your research at: www.wileyauthors.com/eo/promotion.

EDITOR COMMENTS

Reviewing Editor:

Thank you for engaging with this productive review process.

Senior Editor:

Thank you for attending to Reviewer 2's remaining concerns, and for acknowledging the limitations of your study. I am pleased to report that your manuscript is now considered acceptable for publication in The Journal of Physiology, and thank you for your contribution.

REFEREE COMMENTS

Referee #2:

Thank you for the clarification.